# Differential PROTAC substrate specificity dictated by orientation of recruited E3 ligase

Blake E. Smith[1], Stephen L. Wang [1], Saul Jaime-Figueroa[1], Alicia Harbin[2], Jing Wang[2], Brian D. Hamman[2] & Craig M. Crews [1,3,4]

PROteolysis-TArgeting Chimeras (PROTACs) are hetero-bifunctional molecules that recruit an E3 ubiquitin ligase to a given substrate protein resulting in its targeted degradation. Many potent PROTACs with specificity for dissimilar targets have been developed; however, the factors governing degradation selectivity within closely-related protein families remain elusive. Here, we generate isoform-selective PROTACs for the p38 MAPK family using a single warhead (foretinib) and recruited E3 ligase (von Hippel-Lindau). Based on their distinct linker attachments and lengths, these two PROTACs differentially recruit VHL, resulting in degradation of p38α or p38δ. We characterize the role of ternary complex formation in driving selectivity, showing that it is necessary, but insufficient, for PROTAC-induced substrate ubiquitination. Lastly, we explore the p38δ:PROTAC:VHL complex to explain the different selectivity profiles of these PROTACs. Our work attributes the selective degradation of two closely-related proteins using the same warhead and E3 ligase to heretofore underappreciated aspects of the ternary complex model.

[1] Department of Molecular, Cellular, and Developmental Biology, Yale University, 219 Prospect Street, New Haven, CT 06511, USA. [2] Arvinas, Inc., 5 Science Park, New Haven, CT 06511, USA. [3] Department of Chemistry, Yale University, New Haven, CT 06511, USA. [4] Department of Pharmacology, Yale University, New Haven, CT 06511, USA. Correspondence and requests for materials should be addressed to C.M.C. (email: craig.crews@yale.edu)

A severe limitation of traditional pharmaceutical inhibitors is their dependence on binding affinity to druggable active sites in order to achieve sufficient inhibition[1,2]. Such an approach requires high and sustained doses of inhibitor to reach occupancy-based therapeutic effect and relies on minor differences in highly-conserved enzymatic pockets to attain substrate selectivity[3–7]. Alternatively, PROteolysis-TArgeting Chimeras (PROTACs) have emerged as a promising therapeutic approach that targets proteins to the proteasome for degradation to overcome occupancy-based active site limitations, thereby expanding the potential "druggable proteome"[8–13].

In their simplest form, PROTACs couple a small molecule binder of a target protein (warhead) to an E3 ubiquitin ligase-recruiting moiety via an intervening chemical linker[14–17]. This hetero-bifunctional small molecule design induces the proximity between target protein and E3 ubiquitin ligase complex, promoting the ubiquitination and subsequent degradation of the former[18–20]. To date, efforts have largely focused on recruitment of the von Hippel-Lindau (VHL) and cereblon (CRBN) E3 ubiquitin ligases due to the discovery of high-affinity ligands for these two ligases[21–24]. Endogenously, VHL is the substrate-binding element of the Cullin 2 RING E3 ligase (CRL2$^{\text{VHL}}$ complex), constitutively targeting Hypoxia-inducible factor 1-alpha (HIF-1α) for degradation through a stereospecific interaction with its hydroxyproline motif[25,26]. This intimate HIF-1α: VHL interaction was initially co-opted by our laboratory in 2004[27] and since then has resulted in all-small molecule, VHL-utilizing PROTACs that potently target nuclear, cytosolic, and membrane-bound proteins for degradation[28–32].

Owing to the modular nature of PROTACs, recent work has underscored the importance of warhead selection and E3 ligase choice in the PROTAC-induced degradation of specific proteins. In particular, our group showed that while dasatinib-based CRBN-recruiting PROTACs could degrade both c-Abl and BCR–Abl, dasatinib-based VHL-recruiting PROTACs could degrade only c-Abl;[33] however, when using a bosutinib warhead, VHL-recruiting PROTACs could not degrade either protein target. Additionally, VHL- and CRBN-recruiting PROTACs based on the Bromodomain and Extra-Terminal (BET)-targeting triazolodiazepine JQ1 showed potent degradation of BRD2, BRD3, and BRD4[34] and early characterization of thienopyridinone-based CRBN-recruiting PROTACs demonstrated the additional ability to degrade BRD4, BRD7, and BRD9 proteins[35].

These studies have been contextualized by the publication of a BRD4:PROTAC:VHL crystal structure, in addition to two studies that surveyed proteome-wide degradation using PROTACs based on promiscuous warheads[36–38]. The ternary complex crystal structure revealed key protein-protein interactions (PPIs) between VHL and BRD4, as well as mutual interactions with the PROTAC linker[36,39]. On a proteomic scale, one study demonstrated that ternary complex formation is far more predictive of PROTAC-induced substrate degradation than is target:PROTAC binding affinity[37]. Using the promiscuous kinase inhibitor, foretinib, as a warhead it showed that VHL- and CRBN-utilizing PROTACs resulted in different degradation profiles, but each degradation profile was, similarly, only a subset of the kinases to which each PROTAC bound; further, these PROTAC-bound kinases were themselves more restricted than the body of proteins to which foretinib itself binds. These degraded substrate proteins included even those with only weak binding affinity for the PROTAC (e.g., p38 Mitogen-activated protein kinase [MAPK]), thus demonstrating how compensatory PPIs afforded by the substrate:PROTAC:ligase interface provide an added layer of selectivity and affinity for target proteins[36,37,39].

Unlike previous studies comparing the effectiveness between PROTACs with different warheads and recruiting different E3 ligases, we hypothesized that it would be possible to generate degraders that discriminate between closely-related proteins using only a single warhead and E3 ligase. Starting from our group's previous work[37], we focused on members of the p38 MAPK family because of their lack of isoform-selective chemical probes and their vast and varied roles in disease[40–42]. Consisting of four members (α, β, γ, δ), the p38 MAPK kinases respond to environmental stress and cytokines while displaying differential tissue expression[43,44]. Since its discovery in 1994, p38α has been the best-studied isoform, and while dozens of inhibitors have been developed for clinical trials, none have demonstrated the efficacy and safety needed to receive Food and Drug Administration (FDA) approval[45–47]. Alternatively, p38δ is not only an understudied stress-sensing kinase with roles in cancer and diabetes, but its constrained ATP-binding pocket has made discovery of selective inhibitors difficult[48–51]. In fact, a recent study using a library of 178 commercially available kinase inhibitors with selectivity for every major protein kinase subfamily suggested that p38δ was altogether intractable to functional inhibition[52].

Here, we develop p38α- and p38δ-selective PROTACs based on a single warhead (foretinib) and E3 ligase (VHL), and allow differences in linker length and orientation of the recruited VHL to bias the degradation outcome of closely-related kinase isoforms. We find that formation of a ternary complex is a necessary but insufficient step in discriminating between substrate proteins and determine that the affinities of these interactions are highly indicative of cellular outcomes.

## Results

**Identification of p38 isoform-selective PROTACs.** We began to address these mechanistic questions surrounding PROTAC substrate specificity by synthesizing eight foretinib-based PROTACs that vary by linker length and by orientation of the VHL-recruiting molecule. The first four PROTACs (termed the "amide series") were constructed using the previously characterized left-hand side amide[29] as a connection point for linker attachment to the VHL ligand, whereas the remaining four PROTACs (termed the "phenyl series") incorporated an under-explored right-hand side phenyl ring[28] attachment point of the VHL ligand to the linker (Figs. 1a, b). Both PROTAC series utilized the same phenyl ether attachment from the solvent-exposed region of the foretinib warhead and used linker lengths of 10, 11, 12, and 13 atoms (Supplementary Figures 1-2). We reasoned that PROTACs synthesized with different linker lengths, using two different points of attachment to the VHL-recruiting ligand, could promote distinct E3 ligase interfaces with a given p38 isoform, potentially enabling degradation selectivity.

In consideration of the "amide series", we discovered that the 10-atom and 11-atom linker PROTACs (SJF-6693 and -6690) are not selective and degrade both p38α and p38δ with DC$_{50}$ < 100 nM in MDA-MB-231 human breast cancer cells (Supplementary Table 1, Supplementary Figures 1-2). However, this promiscuity became drastically more p38α-selective with the 12-atom (SJF-8240) and 13-atom (SJFα) linker PROTACs. SJFα degrades p38α with a DC$_{50}$ of 7.16 ± 1.03 nM and $D_{\text{max}}$ of 97.4%, but is far less effective at degrading p38δ ($D_{\text{max}}$ = 18%, DC$_{50}$ = 299 nM) and does not degrade the other p38 isoforms (β and γ) at concentrations up to 2.5 μM (Figs. 1c, d, Supplementary Figures 3-4, Supplementary Table 1). Conversely, the longer linker-based PROTACs in the "phenyl series" demonstrated little degradation of any p38 isoform. However, shortening the linker length in the "phenyl series" down to 10 atoms produced a PROTAC with strong capacity to degrade p38δ selectively: SJFδ degrades p38δ with a DC$_{50}$ of 46.17 ± 9.85 nM and a $D_{\text{max}}$ of 99.41 ± 3.31% (Figs. 1c, d) but does not degrade p38α, β, or γ (Supplementary

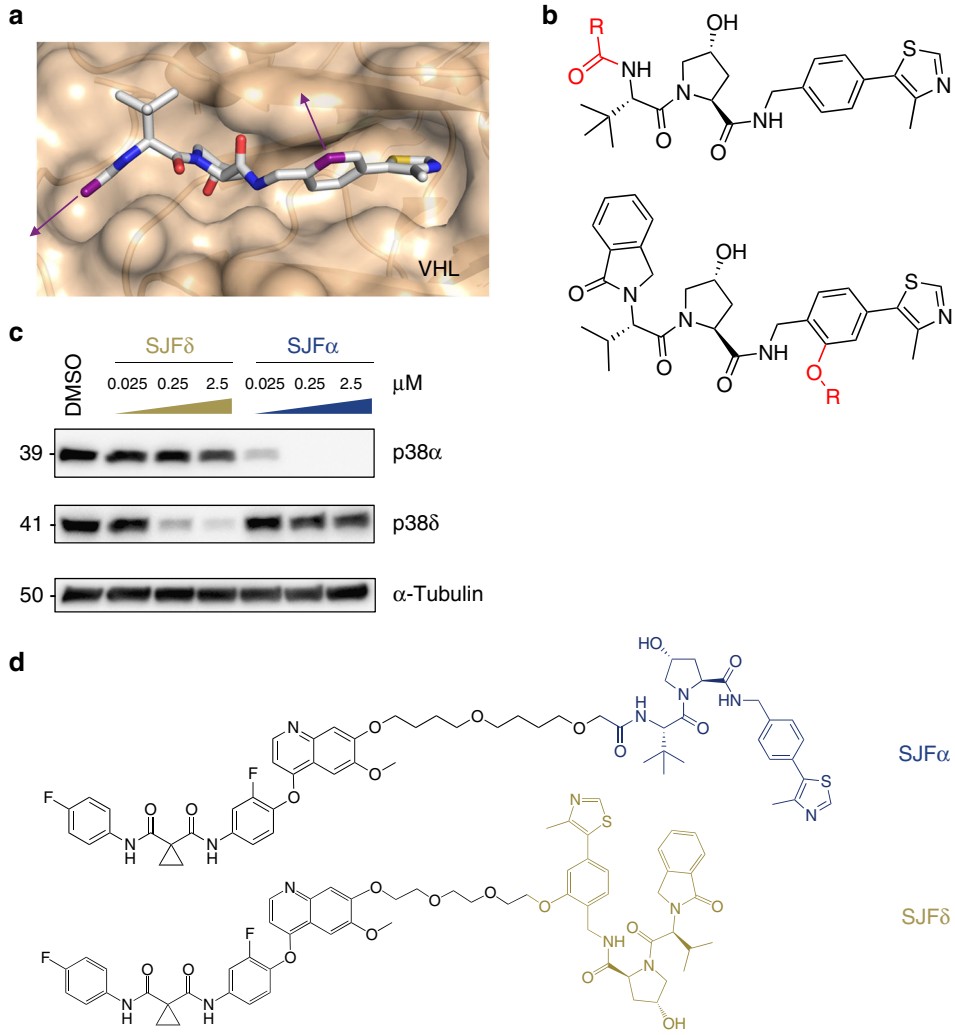

**Fig. 1** Foretinib-based PROTACs recruit VHL using two different linkage vectors resulting in isoform-selective degradation. **a** Crystal structure (PDB: 4W9H) of VHL-recruiting ligand in the HIF-1α-binding pocket. VHL is rendered as a tan surface with the small molecule ligand visualized in stick representation and colored by atom (gray carbon atoms, red oxygen atoms, blue nitrogen atoms, and yellow sulfur atoms), with the exception of two carbon atoms colored in purple. These purple carbons (and the corresponding purple arrows) represent the structure-guided linker attachment points used in subsequent PROTAC design (see Supplementary Figure 1). **b** Structures of the two VHL-recruiting ligands used in this study. Top: VHL-recruiting ligand with an amide attachment vector (shown in red). Bottom: VHL-recruiting ligand with a phenyl attachment vector (shown in red). **c**, **d** SJFα PROTAC (13-atom linker, amide attachment) selectively degrades p38α, whereas SJFδ PROTAC (10-atom linker, phenyl attachment) selectively degrades p38δ in MDA-MB-231 cells. See Supplementary Figure 1 for the structures of all of the PROTACs screened in this study with their corresponding western blots (Supplementary Figures 2-3) and summary table (Supplementary Table 1). See Supplementary Figure 4 for further investigation into the activity of the lead PROTACs on p38, ERK, and JNK MAPK families

Figures 2-4, Supplementary Table 1). In certain instances, the change in linker length by even a single carbon atom alters the degradation selectivity observed to a striking degree. For example, although SJF-8240 of the "amide series" with its 12-atom linker displayed sub-micromolar potency to degrade both p38 isoforms, lengthening the linker by 1 atom to create SJFα increased potency to degrade p38α into the low nanomolar range while simultaneously decreasing p38δ degradation potency to the micromolar range (dramatically reducing efficacy as well). Likewise, SJFδ – of the "phenyl series" with its 10-atom linker – degrades p38δ to near-completion but is limited in degrading p38α; by comparison, SJF-6677, with an additional carbon in its linker degrades <50% of either isoform at maximum effectiveness. Moreover, there was no hook effect[53] observed up to 100 μM with either SJFα or SJFδ (Supplementary Figure 3) and neither PROTAC degrades the related MAPKs extracellular-signal-regulated kinase 1 (ERK1), ERK2, c-Jun N-terminal kinase 1 (JNK1), and JNK2

(Supplementary Figure 4), demonstrating the profound isoform selectivity of our two lead PROTACs. Having identified these lead compounds, we moved to investigate how differential orientation of recruited VHL, coupled with different PROTAC linker lengths, produced two PROTACs that can each degrade a single p38 MAPK isoform.

**PROTACs achieve proteasome-mediated p38 isoform degradation.** We sought to confirm that this intriguing isoform-selective loss of p38 observed was due only to post-translational degradation. PROTACs function through degradation of existing target proteins via the ubiquitin proteasome pathway, rather than by suppression of target protein synthesis (as is the case with nucleic acid-based knockdown strategies). Accordingly, we investigated whether SJFα- and SJFδ-mediated downregulation of their respective targets proceeds in a manner consistent with the

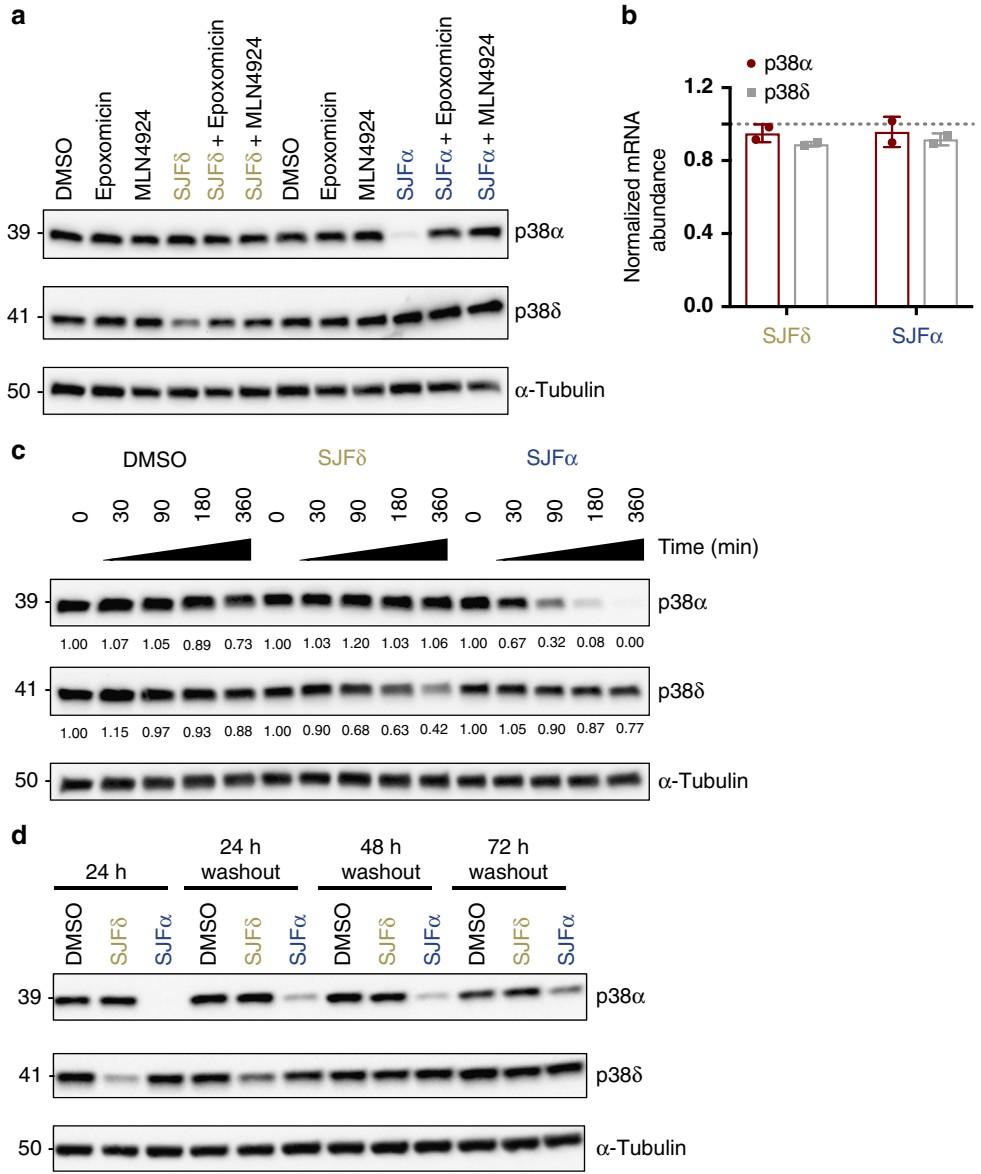

**Fig. 2** SJFα and SJFδ degradation of p38 isoforms is rapid, sustained, and proteasome dependent. **a** MDA-MB-231 cells were either pre-treated or not with the proteasome inhibitor epoxomicin (1 μM) or the neddylation inhibitor MLN4924 (1 μM) for 30 min and subsequently treated with either vehicle (DMSO), SJFα (250 nM), or SJFδ (250 nM) for 6 h. **b** Quantitative real-time PCR was performed after 24 h of treatment with either vehicle (DMSO), SJFα (250 nM), or SJFδ (250 nM) in MDA-MB-231 cells. The dotted line represents DMSO-treated conditions standardized to a value of 1.0 and mRNA abundance (per treatment group) is represented as a fold-change relative to that value. Data are based on biological duplicates and are normalized to beta-tubulin. Error bars denote s.d. **c** Cycloheximide (CHX) chase assay. MDA-MB-231 cells were pre-treated with 100 μg mL$^{-1}$ CHX for 1 h prior to treating for the indicated times with either vehicle (DMSO), SJFα (250 nM), or SJFδ (250 nM). Tubulin-normalized p38α and p38δ abundance values are reported beneath individual lanes. **d** MDA-MB-231 cells were treated with either vehicle (DMSO), SJFα (250 nM), or SJFδ (250 nM) for 24 h before re-plating onto new plastic in fresh medium for an additional 24, 48, or 72 h ("washout" conditions)

mechanism of action of PROTACs. Cells treated with 250 nM SJFα or SJFδ for 6 h resulted in downregulation of p38α and p38δ, respectively, which is completely rescued when cells were pre-treated with 1 μM epoxomicin (proteasome inhibitor)[54] or 1 μM MLN4924 (NEDD8-activating enzyme inhibitor)[55], indicating that these PROTACs depend on the proteasome and ubiquitination cascade (i.e., neddylated CUL2) for their action (Fig. 2a). Furthermore, quantitative real-time PCR performed on PROTAC-treated cells revealed no significant changes in mRNA for either p38 isoform, demonstrating that these PROTACs do not downregulate p38α and p38δ at the transcriptional level (Fig. 2b). Additionally, MDA-MB-231 cells were pre-treated with 100 μg mL$^{-1}$ of cycloheximide (CHX) and chased with 250 nM

PROTAC for 30–360 min, revealing rapid degradation half-lives of SJFα on p38α and SJFδ on p38δ (Fig. 2c). Taken together, these results revealed that neither PROTAC substantially down-regulates p38 either before or at the level of de novo synthesis (since CHX stalls protein synthesis), indicating bona fide post-translational selectivity for their respective p38 isoform. Finally, we sought to monitor the duration of effect of SJFα and SJFδ upon washout, as PROTACs have been shown to exhibit a cat-alytic mechanism of action[29]. MDA-MB-231 cells treated for 24 h with either SJFα or SJFδ were rinsed with phosphate-buffered saline (PBS; to remove excess extracellular compound), dis-sociated from culture dishes, and re-plated onto new culture plates with fresh medium devoid of any compound ("washout"

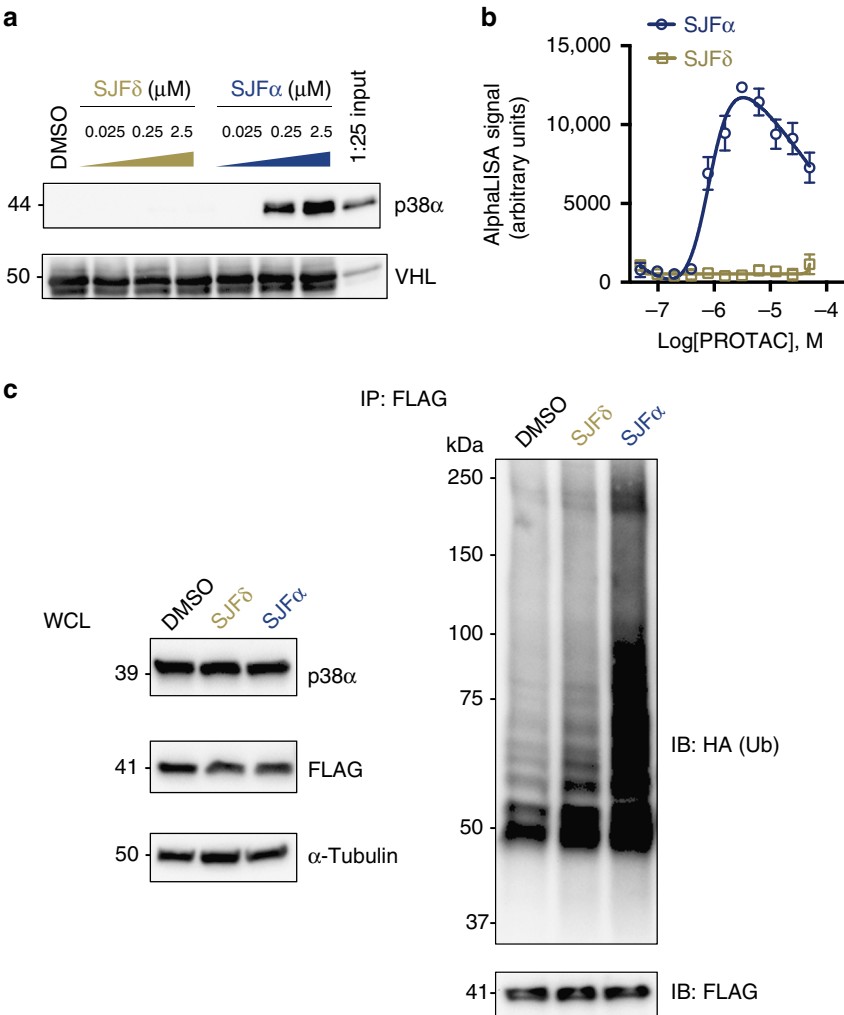

**Fig. 3** p38α selectivity is discriminated by ternary complex formation. **a** Only p38α incubated in the presence of SJFα can be immunoprecipitated with GST-tagged VHL/EloB/EloC (VBC). Immobilized VBC was used as a "bait" to trap purified p38α in the presence of vehicle (DMSO), SJFα, or SJFδ (represented in micromolar concentrations). The rightmost lane represents a 1:25 dilution of initial input protein used in each pull-down. **b** Proximity-based AlphaLISA assay detects significant p38α:SJFα:VHL ternary complex, but no p38α:SJFδ:VHL ternary complex. His-p38α and GST-VBC were incubated in the presence of increasing concentrations of SJFα and SJFδ and the extent of ternary complex formation was assessed by excitation with incident light with $\lambda = 680$ nm and capture of the emission light at $\lambda = 615$ nm. Error bars represent the s.d from quadruplicate experiments. **c** Only SJFα ubiquitinates FLAG-p38α in HeLa cells. HeLa cells co-transfected with HA-Ubiquitin (HA-Ub) and FLAG-p38α were subsequently treated with either vehicle (DMSO), 500 nM SJFα, or 500 nM SJFδ for 1 h. FLAG-immunoprecipitated lysates were separated by sodium dodecyl sulfate polyacrylamide gel electrophoresis (SDS-PAGE) and assessed via western blots detecting HA (Ub). Smears represent ubiquitin-conjugated FLAG-p38α and numerical markers to the left of the western blot refer to kilodalton (kDa) masses. WCL refers to "whole-cell lysate" input

cells). Sustained PROTAC-induced p38 degradation was still observed (Fig. 2d): in particular, SJFα maintained p38α degradation efficacy for 72 h post-washout, whereas SJFδ maintained p38δ degradation efficacy for only 24 h post-washout. These results indicate that our two lead PROTACs degrade their respective p38 isoform in a rapid, sustained, and proteasome-dependent manner.

**p38α degradation selectivity determined by ternary complex.** Previous work has demonstrated the profound influence of favorable PPIs – that occur at the PROTAC-induced interface between a target protein and recruited E3 ligase – on PROTAC degradation outcomes[36,37]. One study of ours revealed that the weak binary affinity of a foretinib-based PROTAC for p38α can be overcome by cooperative binding contributions between VHL and p38α to form a higher affinity ternary p38α:PROTAC:VHL complex[37]. Following this logic, we sought to examine the

selectivity of SJFα for p38α at the ternary complex level using an in vitro ternary complex pull-down assay. Briefly, glutathione S-transferase (GST)-tagged VHL/Elongin B/Elongin C (VBC) "bait" was immobilized on glutathione sepharose beads to trap any potential ternary complexes occurring between VHL and substrate proteins when incubated with a PROTAC. Using this GST-VBC "bait", we observed substantial enrichment of purified p38α only in the presence of SJFα and no detectable enrichment at any concentration of SJFδ tested (Fig. 3a). Similarly, we evaluated the lead PROTACs in a proximity-based luminescence assay (AlphaLISA) to corroborate ternary complex formation. Only SJFα facilitates a VHL:PROTAC:p38α ternary complex, whereas no such ternary species is detectable when p38α and VHL are incubated with increasing concentrations of SJFδ (Fig. 3b).

Following these observations, we reasoned that only SJFα would induce substantial ubiquitination of p38α in a cellular system. Indeed, when HeLa cells co-expressing FLAG-tagged p38α (~ 42 kDa) and HA-ubiquitin (HA-Ub, ~ 10 kDa) were

treated with vehicle (dimethyl sulfoxide [DMSO]), SJFδ, or SJFα for 1 h, immunoprecipitated p38α from only the SJFα-treated cells displayed substantial, high molecular weight (HMW) poly-Ub conjugation (HA-Ub smear) (Fig. 3c); this effect was not seen in the immunoprecipitated p38α from the DMSO- and SJFδ-treated cells, although those samples revealed some p38α mono-, di-, and tri-Ub conjugates. Both the signal intensity and size of the SJFα-induced poly-Ub chains (up to ~ 100 kDa) indicate a p38α protein with up to ~ 6-7 ubiquitin proteins attached (Fig. 3c). These SJFα-induced p38α-Ub conjugates correlate with the substantial p38α:SJFα:VHL ternary complex observed (Figs. 3a, b), in addition to the rapid, sustained, and selective degradation of p38α by SJFα and not SJFδ (Figs. 1, 2).

**Ternary complex differences select for p38δ degradation.** Upon discovering that SJFα selectivity for p38α occurs at the level of the ternary complex, we similarly examined SJFδ selectivity for p38δ. Using the in vitro ternary complex assay, we identified that both SJFα and SJFδ can pull-down purified p38δ on GST-VBC-coated beads in a dose-dependent manner, with SJFδ showing only slightly greater ternary complex pull-down efficiency than SJFα at all concentrations tested (Fig. 4a). This curious finding – that non-degraded proteins can engage in PROTAC-induced ternary complexes – was recently reported[37], although not compared with PROTACs that promote degradation of those same targets. We sought to explore this intriguing possibility whereby two PRO-TACs that can engage with p38δ in a ternary complex in vitro result in different cellular outcomes: target degradation (SJFδ) or not (SJFα). Thus, we characterized the p38δ:PROTAC:VHL ternary complexes using surface plasmon resonance (SPR) to quantify binary and ternary complex affinities and assessed the relative contributions of assembly kinetics in dictating these affinities.

In the SPR experiments, parent warhead foretinib, SJFα, and SJFδ were injected onto immobilized p38δ (His-tagged p38δ) in separate channels to obtain binary p38δ:compound affinity measurements (Supplementary Figure 5a). Foretinib displays an affinity of 240 nM – in agreement with previous measurements[56] – and SJFα ($K_d = 500$ nM) and SJFδ ($K_d = 550$ nM) show about twofold loss of affinity for p38δ compared with the warhead. Using immobilized VHL (GST-VBC), we obtained binding affinities for VHL:SJFα ($K_d = 5.0\ \mu M$) and VHL:SJFδ ($K_d = 3.5\ \mu M$) (Supplementary Figure 5b). Using the same setup, equimolar p38δ:PROTAC dilutions were successively injected onto immobilized VHL for 60 s, prior to monitoring complex dissociation for the remaining 300 s. This SPR strategy enabled us to determine ternary affinities for a given p38δ:PROTAC:VHL complex and compare the relative affinity change that occurs between VHL:PROTAC when p38δ is present. As shown in Table 1, the observed p38δ:SJFδ:VHL affinity is 436 nM, which represents an ~ 8-fold leftward shift – nearly an order of magnitude affinity improvement – from the VHL:PROTAC binary affinity. This enhanced composite affinity is likely due to the additional PPIs between VHL and p38δ formed in the presence of SJFδ[53]. SJFα, however, displayed a p38δ:SJFα:VHL affinity of 1.2 μM, which represents a small improvement over the cognate binary affinity, but is nevertheless ~ 3-fold weaker than the SJFδ-induced complex. From a kinetics perspective, both p38δ:SJFδ and p38δ:SJFα displayed rapid $k_{on}$ kinetics of $3.96 \times 10^4\ M^{-1}\ s^{-1}$ and $7.01 \times 10^4\ M^{-1}\ s^{-1}$, respectively, and both complexes displayed slow dissociation rates. However, the SJFδ complex ($k_{off} = 0.018\ s^{-1}$) showed increased half-life ($t_{1/2} = 38$ s), compared with the SJFα complex ($k_{off} = 0.072\ s^{-1}$, $t_{1/2} = 8$ s) (Table 1 and Supplementary Figure 5c). These results indicate that p38δ:SJFδ:VHL is the more favorable, highly-populated ternary

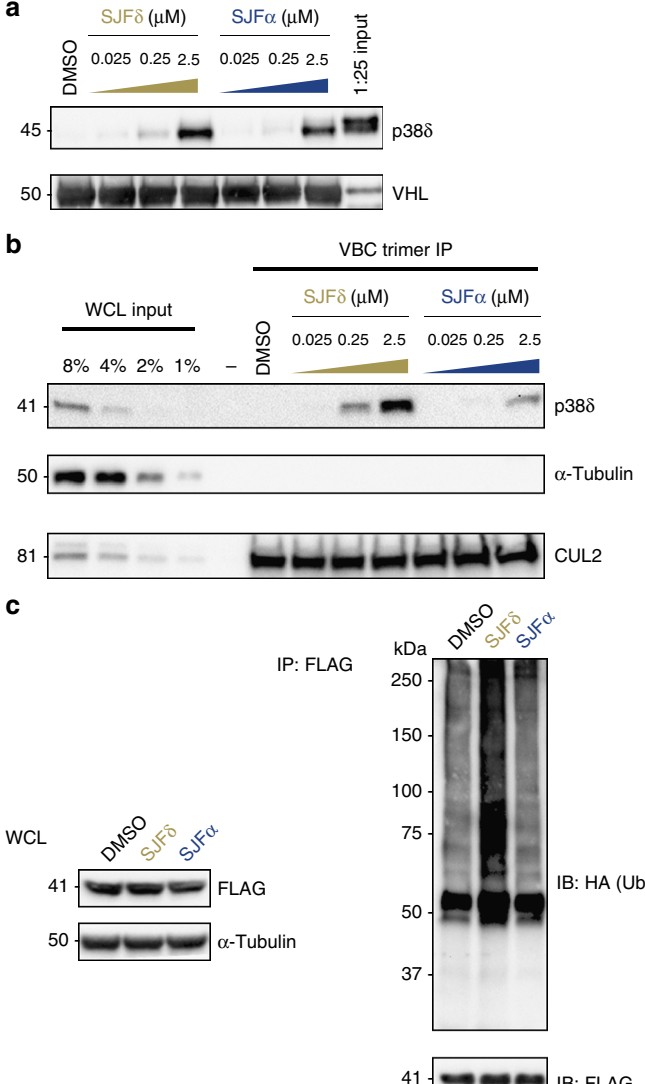

**Fig. 4** Differential p38δ:PROTAC:VHL complex affinities result in distinct cellular ubiquitination. **a** Both SJFα and SJFδ pull-down p38δ in vitro, however, SJFδ maintains ternary complex at lower concentrations. As in Fig. 3a, GST-tagged VBC was used as a "bait" to trap recombinant p38δ in the presence of vehicle (DMSO) or the indicated concentrations of SJFα or SJFδ. The rightmost lane represents a 1:25 dilution of initial input protein used in each pull-down. **b** SJFδ forms a more stable ternary complex with endogenous p38δ and VHL than does SJFα. GST-VBC was immobilized on glutathione sepharose beads and incubated with MDA-MB-231 whole-cell lysate (WCL) in the presence of vehicle (DMSO) or the indicated concentrations of SJFα or SJFδ. Beads were washed and "trapped" proteins (i.e., those that engage in a ternary complex) were eluted with SDS sample buffer and separated by SDS-PAGE. Samples were assessed by western blot and CUL2 and α-tubulin serve as positive and negative GST-VBC immunoprecipitation controls, respectively. **c** Only SJFδ ubiquitinates FLAG-p38δ in HeLa cells. As in Fig. 3c, HeLa cells were co-transfected with HA-Ub and FLAG-p38δ and were either treated with vehicle (DMSO), 1 μM SJFα, or 1 μM SJFδ for 2 h. FLAG-immunoprecipitated lysates were separated by SDS-PAGE and assessed via western blots detecting HA (Ub). Smears represent ubiquitin-conjugated FLAG-p38δ and numerical markers to the left of the western blot refer to kilodalton (kDa) masses. See Supplementary Figures 5 and 6 for additional p38δ:PROTAC:VHL characterization

**Table 1 Summary of binary (PROTAC:p38δ or PROTAC: VHL) and ternary (p38δ:PROTAC:VHL) affinity and kinetic measurements**

|  | SJFδ | SJFα |
| --- | --- | --- |
| PROTAC:p38δ $K_d$ | 550 nM | 500 nM |
| PROTAC:VHL $K_d$ | 3500 nM | 5000 nM |
| p38δ:PROTAC:VHL $K_d$ | 436 nM | 1200 nM |
| p38δ:PROTAC:VHL $k_{off}$ | 0.018 s$^{-1}$ | 0.072 s$^{-1}$ |
| p38δ:PROTAC:VHL $k_{on}$ | $3.96 \times 10^4$ M$^{-1}$s$^{-1}$ | $7.01 \times 10^4$ M$^{-1}$s$^{-1}$ |
| p38δ:PROTAC:VHL $t_{1/2}$ | 38 s | 8 s |

The last protein listed in each list above represents that which was immobilized on the SPR instrument. See Methods section for further descriptions

complex of the two, correlating with the degradation outcomes seen (Fig. 1c, Supplementary Figure 3).

To validate these findings, we used a more biologically relevant context for measuring ternary complex formation. We employed two strategies utilizing MDA-MB-231 whole-cell lysate (WCL) as the source for both endogenous p38δ and VHL, in addition to any associated proteins. In the first approach, we used a modified ternary complex pull-down assay in which WCL from MDA-MB-231 cells was incubated with vehicle (DMSO) or varying concentrations of either SJFα or SJFδ in the presence of excess immobilized VHL (GST-VBC) bait. We found that SJFδ strongly enriches for endogenous p38δ in a dose-dependent manner while stable ternary complex formation was greatly reduced in the SJFα-incubated samples (Fig. 4b). In the second strategy, we utilized a cellular thermal shift assay (CETSA), which recently showed that PPIs in cyclin-containing protein complexes enhance a substrate's thermal stability when subjected to a range of denaturation (melting) temperatures[57,58], to explore whether PROTAC-induced PPIs could be monitored in a complex cellular environment[59]. Previously, we determined that free (unbound) endogenous p38δ in MDA-MB-231 cells displays half-maximal melting at 53 °C and accordingly chose to center our experiments around this temperature. Upon incubation with SJFα, a small, but statistically significant ($p = 0.0121$, Student's $t$-test) thermal shift in the p38δ melting profile was observed, indicating limited target engagement between p38δ and SJFα (Supplementary Figure 6). However, the thermal stability shifts in p38δ were more pronounced at each melting temperature when cells were incubated rather with SJFδ and showed statistical-significance ($p < 0.0001$, Student's $t$-test) compared with DMSO. Notably, thermal shifts were not seen comparing SJFα or SJFδ with DMSO on the negative control protein (α-tubulin), indicating bona fide PROTAC-induced p38δ thermal stabilization. We reasoned that these SJFδ-induced and SJFα-induced thermal stability shift differences are not likely due to differences in cellular uptake nor binary affinity, as these CETSA assays were performed with MDA-MB-231 cell lysate and SJFα possesses slightly greater binary affinity for p38δ than does SJFδ (Table 1). Instead, we believe the increased SJFδ-induced p38δ thermal stability is due to its increased ternary association with VHL (Fig. 4b, Table 1, Supplementary Figure 5b-c), compared with SJFα. Moreover, the differences seen between these two assays and Fig. 4a demonstrate how the more rigorous assessment of ternary complex formation in the cellular milieu might be required, especially for ternary complexes with weak affinity (p38δ:SJFδ:VHL, see Discussion). Thus, much like our previous study[37], this p38δ:SJFδ:VHL "cellular ternary complex" strongly correlates with the selective p38δ degradation observed with SJFδ in MDA-MB-231 cells (Fig. 1c), and we posit that the mere capacity to form a ternary complex is insufficient for PROTAC-induced substrate degradation.

Based on these PROTAC-induced ternary complex affinity differences, both in vitro and in cells, we questioned whether they were predictive of PROTAC-induced p38δ cellular ubiquitination. When HeLa cells co-expressing FLAG-p38δ (~ 43 kDa) and HA-Ub were treated with vehicle (DMSO) or saturating concentrations of either SJFα (1 μM) or SJFδ (1 μM), only the SJFδ-treated cells displayed levels of ubiquitination of immunoprecipitated p38δ greater than control cells (Fig. 4c). Substantial p38δ-Ub conjugates are seen in the 75–100 kDa region where tetra-ubiquitinated[60,61] p38δ is expected to migrate, but these are far less abundant in the DMSO- and SJFα-treated samples. Similar experiments performed with agarose beads conjugated to tetra-ubiquitin-capturing tandem ubiquitin binding entities (TUBEs)[62] revealed that only SJFδ-treated cells display increased TUBE1-reactivity (poly-Ub p38δ, Supplementary Figure 7). Taken together, these results indicate that SJFα and SJFδ ternary complex affinity differences, in vitro and in cells, result in profoundly different cellular ubiquitination profiles that may explain the p38δ degradation selectivity of SJFδ (Fig. 1c, Supplementary Figure 3).

To investigate the possible structural bases for the isoform-specific ternary complex affinity differences, we performed molecular dynamics (MD) simulations on the p38δ:SJFδ:VHL and p38δ:SJFα:VHL complexes. We docked previously published p38δ and VHL structures and allowed these two ternary structures to separately relax into a low energy conformation. These short (100-ns) MD simulations showed that SJFα and SJFδ result in a recruited VHL that docks onto p38δ in two entirely different conformations (Fig. 5a). According to these models, the pentameric E3 ligase complex (VHL/EloB/EloC/CUL2/Rbx1) accesses different faces of p38δ owing to the two contorted VHL conformations seen in the SJFα-recruited and SJFδ-recruited ternary complexes. Moreover, upon inspecting the conformation of each PROTAC in these models, we found that the 10-atom phenyl-linked SJFδ directly interacts with VHL outside the p38δ kinase pocket, whereas the 13-atom amide-linked SJFα takes a turn out of the p38δ kinase pocket, kinking along the way, to recruit a VHL that results in a twisted conformation, relative to the SJFδ-recruited p38δ (Fig. 5b). These comparative models highlight how linker length and orientation of the recruited E3 ligase can result in different ternary interfaces (with inherent ternary complex affinity differences) to achieve selective degradation with one PROTAC over another.

To further investigate the differences between the two MD simulations, we compared p38δ and VHL PPIs in the presence of the degrader SJFδ (Fig. 6a) to the predicted PPIs that would occur between these same proteins in the presence of SJFα, the non-degrading PROTAC (Fig. 6b). Our model predicted that Arg108 in VHL makes unfavorable interactions with a pair of amino acids in p38δ – Lys220 and Thr221 – in the presence of SJFα; whereas in the presence of SJFδ, this same Arg108 makes stabilizing interactions with a different pair of amino acids, Glu49 and Glu160. Curiously, this electrostatic arm of Arg108 interacts with Asp residues in HIF-1α and Asp/Glu residues in BRD4 (in the recently reported PROTAC crystal structure, PDB: 5T35 [36]) to stabilize their interaction with VHL. To verify a similar role of this residue in our system, we performed mutagenesis of p38δ to see whether substitution of Lys220 and Thr221 with adjacent Glu residues might permit a more favorable interaction of p38δ with Arg108 of VHL in the presence of SJFα, effectively mirroring the stabilizing interaction predicted between p38δ and VHL in the presence of SJFδ. Crucially, mutation of K220 and T221 to Glu residues in p38δ were not predicted to be disruptive to the p38δ:SJFδ:VHL ternary complex. We were encouraged to find that, when expressed in HeLa cells and tested with either PROTAC, both

**a**

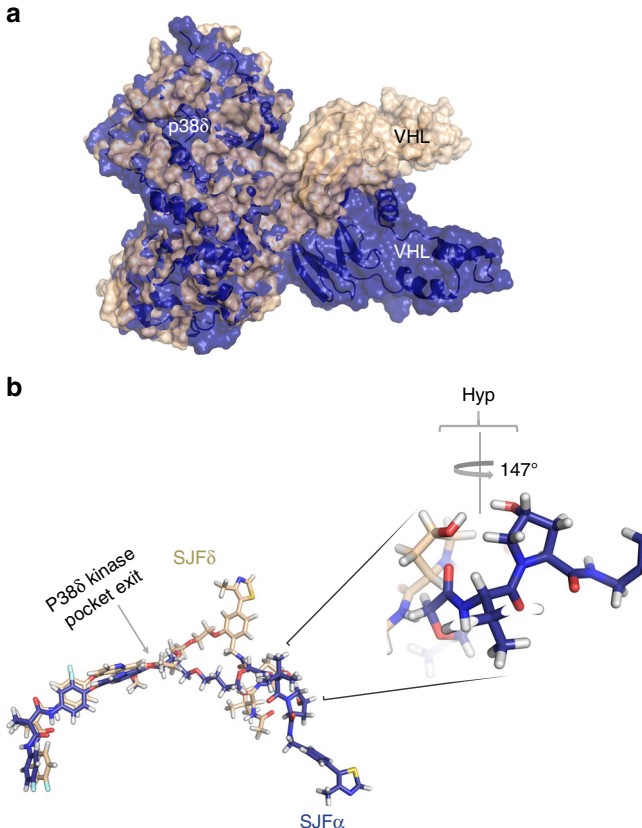

**b**

Fig. 5 MD simulations reveal PPI interfaces between p38δ and VHL that vary between SJFδ and SJFα recruitment. **a** The SJFδ-recruited VHL interacts with p38δ in a different mode than the SJFα-recruited VHL. p38δ, VHL, and either SJFδ or SJFα were docked and a 100-ns MD simulation relaxed the ternary structure. The p38δ:SJFδ:VHL ternary complex is colored in tan and the p38δ:SJFα:VHL ternary complex is colored in dark blue. The p38δ structures from both ternary complex MD simulations were aligned and the resulting divergent VHL structures can be visualized. **b** SJFδ and SJFα PROTACs deviate at the exit of the p38δ kinase pocket to interact with the p38δ:VHL PPI interface in distinct ways. Using the same p38δ alignment as in (**a**), the p38δ and VHL structures were hidden such that individual PROTACs could be visualized in stick representation. SJFδ carbons are colored in tan and SJFα carbons are colored in dark blue with oxygen atoms colored in red, nitrogen atoms in blue, sulfur atoms in yellow, and fluorine atoms in pale blue for both PROTACs. SJFδ or SJFα (which are composed of the same foretinib warhead) show profound alignment within the p38δ kinase, but deviate greatly upon exiting the pocket. Linker length differences between SJFδ (10 atoms) and SJFα (13-atoms) reveal the limits in their respective ability to dock along the p38δ:VHL PPI interface. In addition, the orientation of attachment of the VHL-recruiting ligands result in hydroxyproline (Hyp) moieties that differ by ~147° between the two PROTAC structures

SJFδ or SJFα were each able to degrade the double-mutant p38δ (Figs. 6c, d) with the efficacy of the latter approaching that of the former. Whereas only SJFδ had been able to eliminate wild-type p38δ, selective mutation of those p38δ residues predicted to disrupt SJFα-mediated binding of the kinase to VHL significantly diminished the selectivity profile by expanding the substrate capacity for SJFα. This result reinforces the validity of the MD ternary complex models and sheds light on the molecular basis behind the pharmacological differences that fine tune the isoform selectivity of our p38-targeting PROTACs.

## Discussion

In this study, we sought to develop isoform-selective p38 MAPK-targeting PROTACs using only one warhead and one E3 ligase. Our work was motivated by previous studies that showed how the use of different targeting ligands for a common protein substrate, as well as the choice of recruited E3 ubiquitin ligase (CRBN or VHL), result in drastically different degradation profiles for the substrate[33–35]. In the present study, we asked whether it was possible to selectively target one isoform of a protein family over others simply by varying PROTAC linker design and fostering differential orientation of a single recruited E3 ligase all while using a constant warhead. Starting with the kinase inhibitor foretinib, which exhibits pan-selectivity towards the p38 MAPK family, as the targeting warhead, we developed PROTACs – SJFα and SJFδ – that differentially recruit VHL, and in doing so achieve distinct p38α/p38δ degradation profiles. Through our work, we demonstrated that single-atom alterations to PROTAC linkers could change chemical probes with dual p38α/p38δ degradation activity (SJF-6693 and 6690) into ones with enhanced selectivity for the p38α isoform (SJF-8240 and SJFα) (Supplementary Figure 2, Supplementary Table 1). These amide series PROTACs were based on the oft-utilized "left-hand side" VHL-recruiting moiety and required longer linker lengths to achieve p38α selectivity (SJFα = 13-atom linker). Conversely, only the shortest linker-containing PROTAC of the phenyl series (SJFδ) – which employ a "right-hand side" VHL ligand linkage – demonstrated potent and efficacious activity towards a specific p38 isoform, p38δ. Linker length increases beyond 10 atoms – even single-atom changes – in the phenyl series strongly attenuated their enhanced capacity to degrade p38δ. Although it is possible that VHL might possess an inherent preference for interfacing with a given p38 MAPK isoform, we believe such differences to be nominal due to (i) high structural conservation between the p38 isoforms (Supplementary Figure 4), and (ii) the fact that the shorter amide series PROTACs (10-atom and 11-atom) show similar efficacy to degrade p38α and p38δ. Instead, we believe that these atomic linker length preferences are biased by the VHL-recruiting ligands themselves and selectivity is further enhanced by linker length exploration. Based on the two linkage vectors of the VHL ligand (Figs. 1a, b), we believe that the phenyl attachment provides a more direct VHL recruitment that does not require the PROTAC linker to bend back on itself, as seen in the only PROTAC crystal structure to date (PDB: 5T35), which is based on an amide attachment to the VHL ligand[36]. This kinked linker conformation was also seen in our previous study in which MD simulation of the p38α:PROTAC:VHL ternary complex revealed a residue on p38α (Ala40) that favorably interacted with the amide series PROTAC linker[37]. Although these kinked linkers seen in the amide-linked VHL PROTACs appear to provide additional surface for favorable contact between the substrate:ligase interface, subtle changes to these contacts – either through slight modifications in linker length or composition – appear to result in drastic enhancements/reductions in potency and selectivity (as seen in Supplementary Figure 2, Supplementary Table 1). In fact, the lack of degradation seen with phenyl-linked VHL PROTACs in a previous report[63] might be a consequence of under-explored linker space and the fact that amide-linked and phenyl-linked VHL PROTACs can have different linker length requirements per given substrate. Thus, previous work and the current study both demonstrate that PROTAC linkers represent a delicate balance between affinity contributions and steric effects and that single atom changes to the linker can drastically shift that balance by a yet unresolved mechanism.

In an effort to understand p38 MAPK isoform selectivity, we investigated the ability of the two PROTACs to form a stable

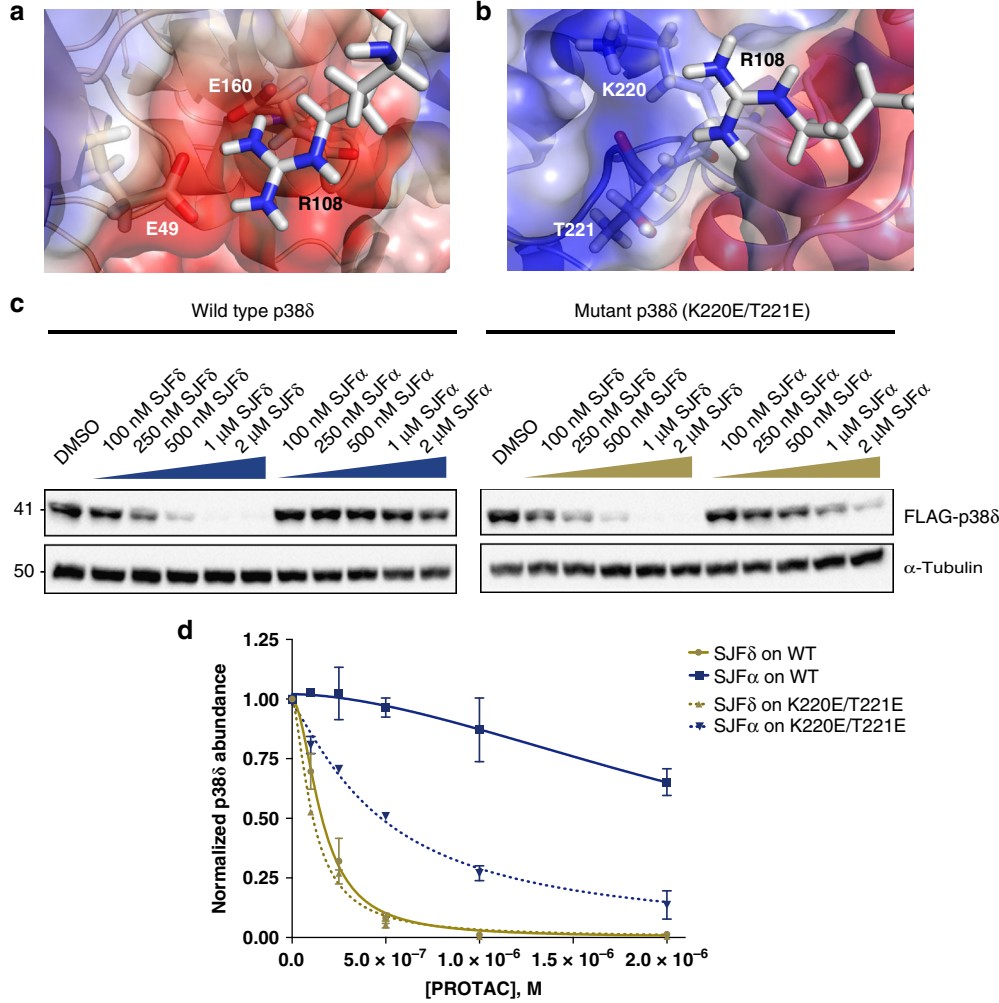

**Fig. 6** Mutation of MD simulation-identified amino acids on p38δ permits SJFα-dependent degradation by VHL. **a** MD simulation identifies amino acid residues of p38δ and VHL that stabilize ternary complex in the presence of SJFδ. p38δ is colored by electrostatic potential, from negative (red) to neutral (white) to positive (blue); interacting residue Arg108 of VHL is shown as sticks against the surface of p38δ interacting with key residues Glu49, Glu160. **b** MD simulation identifies amino acids of p38δ and VHL predicted to disfavor ternary complex in the presence of SJFα. p38δ is colored by electrostatic potential, from negative (red) to neutral (white) to positive (blue); interacting residue Arg108 of VHL is shown as sticks against the surface of p38δ interacting with charged and polar residues K220, T221, respectively. **c** SJFα potently degrades mutated (K220E/T221E) p38δ, but not wild-type p38δ expressed in HeLa cells. **d** Quantitation of wild-type or mutant (K220E/T221E) p38δ levels in transfected HeLa cells treated with either SJFα or SJFδ. Levels of p38δ are normalized to α-tubulin and values expressed relative to those from cells treated with DMSO (vehicle). Error bars display the s.d. of duplicate experiments

ternary complex with either p38α or p38δ, as previous work by our group showed this step was highly predictive for PROTAC-mediated degradation of a given substrate[37]. However, in that study two exceptions to this principle – the kinases c-Abl and Arg – showed profound ternary complex enrichment with a PROTAC that did *not* degrade them. In our current project, we identified how stable ternary complex formation between p38α:PROTAC:VHL drove the selectivity for SJFα-promoted (and against SJFδ-promoted) p38α degradation (Fig. 3). However, we further showed how the mere presence of a ternary complex between p38δ:PROTAC:VHL was not predictive for subsequent degradation, but, rather, a more nuanced understanding was required to explain the SJFδ degradation selectivity for p38δ (Fig. 4). Our findings point to the differences in ternary complex affinity (and the associated metrics of off-rates and half-lives) as the driving force behind SJFδ-selective degradation of p38δ. Additionally, the present study highlights how specific residues (K220, T221) in p38δ disfavor interaction with VHL in the presence of one PROTAC (SJFα), but not another (SJFδ). This initial work

underscores the highly-specific nature of each substrate:PROTAC:ligase interaction and offers insight into the role that stabilizing interactions of three key arginine residues in VHL (Arg69, Arg107, Arg108) play. While their necessity for substrate degradation remains to be proven, their role in promoting both natural and PROTAC-induced substrate degradation is clear[25,26,36,37].

Though the roles that off-rate and half-life play in proper PROTAC function is yet to be uncovered, we recognize that this consideration might be entirely empirical per substrate:ligase pair, as the accessibility of substrate lysines, processivity of a given recruited E3 ligase complex, ubiquitin chain linkage patterns, proteasomal recognition of ubiquitinated lysines, identity of PROTAC-recruited E2 enzymes, and the role that different E2 enzymes play (initiation vs. extension) is yet to be understood. Despite these nuanced considerations, we believe that ternary complex affinity is the main driving force behind SJFδ-mediated selective degradation of p38δ, as these in vitro ternary complex affinities correlate in cells where SJFα engagement with p38δ is

reduced compared with SJFδ (Fig. 4b and Supplementary Figure 5), resulting in a lack of significant cellular ubiquitination of p38δ with SJFα (Fig. 4c and Supplementary Figure 7). We do not believe that lysine accessibility differences between the p38δ:SJFα: VHL and the p38δ:SJFδ:VHL complexes explain selectivity (despite differences in the modeled p38δ:VHL architecture), since p38δ not only possesses twice as many lysines as p38α (32 vs. 16), but these lysines are distributed throughout its surface and 11 of these lysines are homologous between p38δ and p38α (Supplementary Figure 4a–b). Furthermore, we have not ruled out the possibility that minimal degradation of p38δ by SJFα could be due to competition of SJFα:VHL with natural binding partners of p38δ, of which few are known. While no studies have shown competition between PROTAC:VHL and associated proteins of a substrate, this is certainly a possibility. Based on the modeled docking interface of p38δ:PROTAC:VHL (Fig. 5), it appears that VHL binds to a highly-conserved docking groove present in every MAPK family[64–66]. As p38δ is an understudied kinase, few studies previous to this one have characterized p38δ docking groove interactions. However, in p38α this common docking groove – to which upstream kinases, inactivating phosphatases, and downstream substrates bind – displays µM affinities for such interactions[67]. Thus, it is possible that the relative ternary complex affinity differences that occur between p38δ:SJFα:VHL and p38δ: SJFδ:VHL in vitro become magnified in a cellular milieu of docking groove competition. Future studies will investigate whether or not specific target protein-binding partners contribute to unsuccessful PROTAC-induced degradation and whether this hypothesis explains why certain targets are refractory to PROTAC-induced degradation.

Overall, we identify two PROTACs that target individual isoforms of the p38 MAPK family. We achieve selective and potent degradation of p38α (< 10 nM) and p38δ (< 100 nM), for which no FDA-approved therapeutics currently exist despite the vast disease relevance of the former and due to the inhibitor-refractory status of the latter. This study highlights the evolving applicability of the PROTAC technology for selective degradation of protein targets with nM potency through the use of a single E3 ligase and warhead. Furthermore, through exploration of linker length and E3 ligase recruitment geometry, we have underscored that the ternary complex is necessary, but not sufficient, for protein degradation. This study incorporates measurements of PROTAC-induced ternary complex association and dissociation kinetics and comparatively describes the cellular effects of the differences that lie therein. Further work will need to be performed to understand whether the lessons learned here extend beyond the p38 MAPK family, and how the approach could be leveraged to target "undruggable" proteins.

## Methods

**Cell culture**. MDA-MB-231 and HeLa cells were obtained from the American Type Culture Collection (ATCC), cultured in RPMI-1640 medium (1 ×) and Dulbecco's modified Eagle's medium (DMEM; 1 ×), respectively, containing 10% fetal bovine serum and 1% penicillin–streptomycin and grown in a humidified incubator at 37 °C and 5% $CO_2$. For extended dose-response experiments of MDA-MB-231 cells (Supplementary Figure 3), all incubation medium was supplemented with 0.025% Tween 80 and DMSO[68] to enhance PROTAC solubility at the higher concentrations.

**Immunoblotting**. Lysates from MDA-MB-231 cells were washed once with ice-cold PBS (1 ×), followed by lysis in buffer containing 25 mM Tris [pH 7.5], 0.25% sodium deoxycholate, 1% Triton X-100, supplemented with 1X protease inhibitor cocktail (Roche), and phosphatase inhibitors (10 mM NaF, 1 mM Na$_3$VO$_4$, and 20 mM β-glycerophosphate), unless otherwise noted. Lysates were spun at 14,000 × $g$ for 10 min at 4 °C and supernatant was evaluated for protein content using a Pierce BCA Protein Assay (ThermoFisher Scientific). In all experiments, 25–50 µg of protein was loaded onto 10% SDS-PAGE gels or 4–20% Criterion TGX precast gradient gels (Bio-Rad), transferred to nitrocellulose membranes, and probed with the specified antibodies overnight at 4 °C in 1X TBS-Tween (Tris-buffered saline

plus 0.02% Tween 20) containing 5% non-fat milk. Immunoblots were visualized using a Bio-Rad ChemiDoc imaging instrument and subsequently processed and quantified using the accompanying Bio-Rad ImageLab software. See Supplementary Figure 8 for uncropped scans of western blots shown in the main text figures.

Rabbit antibodies purchased from Cell Signaling Technologies (CST) with their respective antibody dilutions are as follows: p38α (#9218) 1:2000, p38 (#2308) 1:1000, p38β (#2339) 1:1000, p38γ (#2307) 1:500, ERK2 (#9108) 1:2000, JNK2 (#9258) 1:1000, and VHL (#68547) 1:5000. Additional antibodies purchased from CST with their respective dilutions are as follows: mouse HA-tag (#2367) 1:1000, mouse JNK1 (#3708) 1:1000, and FLAG (DYKDDDDK) Tag Sepharose bead conjugate (#70569). Mouse antibodies purchased from Sigma-Aldrich with their respective antibody dilutions are as follows: alpha-tubulin (#T9026) 1:5000 and FLAG M2 (#F1804) 1:1000. Rabbit Cullin 2 (CUL2) was purchased from ThermoFisher Scientific (#700179) 1:5000 and rabbit ERK1 (C-16) 1:2000 from Santa Cruz Biotechnology (#sc-93).

**Constructs, protein expression, and purification**. Wild-type human p38alpha-MAPK (MAPK14) was received as a gift from Dr. D. Martin Watterson (North-western University, under MTA) and was described previously[69]. The plasmid contains an N-terminal His$_6$ tag and encodes a region spanning amino acids 2–360 of the human p38α kinase (NCBI Reference Sequence: NM_139012). BL21-CodonPlus(DE3)-RIPL E. coli cells (Agilent Technologies) were transformed with pMCSG7-His$_6$-p38α and were selected in Luria-Bertani (LB) medium containing carbenicillin (100 µg mL$^{-1}$), chloramphenicol (15 µg mL$^{-1}$), and spectinomycin (50 µg mL$^{-1}$) at 37 °C until OD$_{600}$ = 0.6–0.8. At this point, cells were induced with 1 mM isopropyl β-d-1-thiogalactopyranoside (IPTG) and grown at 25 °C for 14–16 h. Cell pellets were collected by centrifugation (5000 rpm, 10 min, 4 °C) and homogenized in lysis buffer (10 mM Tris pH 8.3, 500 mM NaCl, 5 mM β-mercaptoethanol, 10 mM imidazole, and 10% glycerol) containing a 1X protease inhibitor cocktail tablet (Roche). The homogenized cells were subsequently passed through a microfluidizer three times at 15k PSI and lysate was clarified by centrifugation (16,000 rpm, 45 min, 4 °C). The resultant supernatant was then applied to Ni-NTA agarose beads (QIAGEN) with gentle rotation for 1 h at 4 °C, washed once with lysis buffer with 10 mM imidazole (pH 8.3), twice with lysis buffer containing 20 mM imidazole (pH 8.3), and eluted off of the nickel resin in lysis buffer containing 50 mM imidazole (pH 8.3). Eluted protein was assessed for identity and purity via Coomassie staining of sample run on an SDS-PAGE gel and pure elutions were pooled, concentrated, and diluted in ion-exchange buffer A (10 mM Tris pH 8.3, 5 mM β-mercaptoethanol) until the salt concentration was 50 mM, before loading onto a Mono Q 5/50 GL column (GE Life Sciences). The protein was subjected to a step-wise wash protocol, followed by a linear gradient from 200 to 500 mM NaCl using ion-exchange buffer B (10 mM Tris 8.3, 1 M NaCl, 5 mM β-mercaptoethanol). Fractions were then assessed for purity via Coomassie, pooled, concentrated, and run on a HiLoad 16/600 Superdex-200 column (GE Healthcare Life Sciences) using size-exclusion buffer (10 mM Tris pH 8.3, 150 mM NaCl, 5 mM β-mercaptoethanol). Pure fractions of p38α were pooled, concentrated to ~5 mg mL$^{-1}$, aliquoted, and flash-frozen before storing at −80 °C.

Wild-type human p38δ kinase (NCBI Reference Sequence: NM_002754.4) encoding a region spanning amino acids 1–365 was PCR amplified from a pcDNA3.3-p38delta-MAPK (MAPK13) template that we received as a gift from Dr. Romeo Ricci (IGBMC), described previously[70]. This p38δ region was cloned into a pET28a vector containing an N-terminal His$_6$ tag using NheI and BamHI restriction sites. As before, BL21-CodonPlus(DE3)-RIPL E. coli cells (Agilent Technologies) were transformed with pET28a-His$_6$-p38delta and were selected in LB medium containing kanamycin (50 µg mL$^{-1}$), chloramphenicol (15 µg mL$^{-1}$), and spectinomycin (50 µg mL$^{-1}$) at 37 °C until OD$_{600}$ = 0.8. Purification of His$_6$-p38δ was performed identical to His$_6$-p38α above, except Ni-NTA agarose beads were washed twice with lysis buffer with 20 mM imidazole (pH 8.3), once with lysis buffer containing 50 mM imidazole (pH 8.3), and eluted off of the nickel-conjugated resin in lysis buffer containing 150 mM imidazole (pH 8.3). Pure fractions of p38δ were pooled, concentrated to ~ 3 mg mL$^{-1}$, aliquoted, and flash-frozen before storing at −80 °C.

For the expression of GST-tagged VHL:Elongin B:Elongin C (herein referred to as GST-VHL), wild-type human VHL, Elongin B, and Elongin C were co-expressed in E. coli. BL21(DE3) cells were co-transformed with pBB75-Elongin C and pGEX4T-2-VHL-rbs-Elongin B and selected in LB medium containing carbenicillin (100 µg mL$^{-1}$) and kanamycin (25 µg mL$^{-1}$) at 37 °C until OD$_{600}$ = 0.8, at which point the culture was chilled to 16 °C and induced with 0.4 mM IPTG for 16 h. Cells were homogenized and lysed, as described above, except the lysis buffer was composed of 30 mM Tris [pH 8.0], 200 mM NaCl, 5% glycerol, 5 mM DTT containing a 1X protease inhibitor cocktail tablet (Roche). Clarified cell lysate was applied to Glutathione Sepharose 4B beads (GE Life Science) and gently rotated for 2 h at 4 °C. Beads were washed with four column volumes of lysis buffer, followed by four column volumes of elution buffer (50 mM Tris pH 8.0, 200 mM NaCl, 10 mM Glutathione). Eluted protein was assessed for identity and purity via Coomassie staining of sample run on an SDS-PAGE gel and pure elutions were pooled, concentrated, and diluted in ion-exchange buffer A (30 mM Tris pH 8.0, 5% glycerol, 1 mM DTT) until the salt concentration was 50 mM, before loading onto a Mono Q 5/50 GL column (GE Life Sciences). The protein was subjected to a linear gradient of NaCl (0–500 mM NaCl) using ion-exchange buffer B (30 mM

Tris 8.0, 1 M NaCl, 5% glycerol, 1 mM DTT). Fractions were then assessed for purity via Coomassie, pooled, concentrated, and run on a Superdex-200 column (GE Life Sciences) using size-exclusion buffer (30 mM Tris pH 8.0, 100 mM NaCl, 10% glycerol, 1 mM DTT). Pure fractions of GST-VHL were pooled, concentrated to ~5 mg mL$^{-1}$, aliquoted, and flash-frozen before storing at $-80\,°C$.

**Transfections**. Mutant p38δ was generated by QuikChange Lightning site-directed mutagenesis kit (Agilent). Transfections were carried out using Lipofectamine 2000 reagent (Invitrogen) in HeLa cells seeded at $3 \times 10^5$ cells per six well. One microgram of FLAG-containing pcDNA5-p38delta-WT or pcDNA5-p38delta-K220ET221E was used per transfection. Opti-MEM media were changed after 6 h to DMEM ($1 \times$), at which point the indicated compounds were added for 24 h before harvesting.

**Ternary complex pull-down**. Glutathione Sepharose 4B (Glutathione-conjugated beads in a slurry containing 20% ethanol) were washed twice with 1X wash buffer (50 mM HEPES pH 7.5, 150 mM NaCl, 1 mM DTT, 0.01% NP40, 5 mM MgCl$_2$, 10% Glycerol) and then blocked for 1 h at room temperature with 10% bovine serum albumin (BSA) in wash buffer. The beads were then washed again three times with wash buffer and then purified GST-VBC (stable form of VHL complexed with EloB/EloC, described above) was immobilized for 1 h at 4 °C at 3.6 pmole μL$^{-1}$ of beads. The beads were then washed three times with wash buffer, resuspended, divided into two equal volumes, and p38α or p38δ protein was added. The bead:p38 mixture was then aliquoted to separate tubes and PROTAC was added at the indicated concentration (PROTACs were intermediately diluted in 10% DMSO and 0.25% CHAPS). This mixture was incubated at 4 °C for 2 h. The beads were washed three times with 10 column volumes of wash buffer and then eluted with SDS loading buffer at 75 °C for 10 min.

For experiments in which the input substrate is a WCL (Fig. 4b), the sample was prepared as follows. Five 150 mm dishes of confluent MDA-MB-231 cells were washed with 1X PBS, pH 7.4, and then dissociated using enzyme-free PBS-based cell dissociation buffer (ThermoFisher Scientific) for 10 min at 37 °C. Cells were then pelleted, resuspended in wash buffer (same as above, but supplemented with 1X protease inhibitor cocktail (Roche) and 5 μM epoxomicin), and then lysed by sonication (Branson sonicator microtip, power = 6, 50% duty cycle for three cycles of 30 s on and 1 min off at 4 °C). The lysate was cleared by centrifugation and then added to the GST-VBC-conjugated beads as an input substrate, as above.

**AlphaLISA ternary complex assay**. Assays were performed at room temperature and plates sealed with transparent film between addition of reagents to prevent well contamination. All reagents were diluted in 50 mM HEPES pH 7.5, 50 mM NaCl, 69 μM Brij-35, and 0.1 mg mL$^{-1}$ BSA. Recombinant GST-VBC was mixed with His$_6$-p38α and PROTAC (diluted one-in-three from 6× stock) to a final volume of 15 μL per well in a OptiPlate-384 well microplate (PerkinElmer) and incubated for 30 min. VBC and p38α were kept at a constant final concentration of 150 nM. In all, 7.5 μL of Anti-6xHis AlphaLISA Acceptor beads (PerkinElmer) were added to each well and plates were incubated for 15 min in the dark. In total, 7.5 μL of Alpha Glutathione Donor beads (PerkinElmer) were added to each well and plates were incubated for 45 min in the dark. Plates were read on a Synergy 2 microplate reader using the Gen5 imager software (BioTek Instruments) with an excitation wavelength of 680 nm and emission wavelength of 615 nm. Intensity values were plotted in Graphpad Prism with PROTAC concentration values represented on a log10 scale.

**CETSA**. CETSA protocol was adapted from[60]. In all, $3 \times 10^7$ MDA-MB-231 cells ($1 \times 10^7$ cells per condition) were collected and resuspended in ice-cold 1X PBS supplemented with 1X protease inhibitor cocktail (Roche) and lysed by three cycles of liquid nitrogen snap freeze, followed by 50% thaw in a room temperature water bath and an additional 50% thaw at 4 °C. After each freeze–thaw cycle, lysate was vortexed briefly to ensure homogenous thawing. Lysate was then cleared for 20 min by centrifugation ($14,000 \times g$ at 4 °C) and the soluble fraction was divided into three equal aliquots. The aliquots were treated with vehicle (2.5% DMSO), 100 μM SJFα, or 100 μM SJFδ and incubated at room temperature for 30 min with gentle rotation. Each aliquot was then divided (50 μL) into eight PCR tubes and individually heated at the indicated temperature for 3 min followed by cooling at room temperature for 3 min. After cooling, samples were spun down for 20 min ($14,000 \times g$ at 4 °C) and the supernatant (soluble fraction) was analyzed by SDS-PAGE and immunoblotted for p38δ and alpha-tubulin (negative control).

**SPR**. SPR experiments were conducted on a Biacore 3000 instrument (GE Healthcare) at room temperature. GST antibody was immobilized through direct amination onto a carboxymethylated dextran surface (CM5) and GST-VBC was captured on the CM5 surface. This prepared surface was equilibrated over 3 h in running buffer (10 mM HEPES buffer pH 7.4, 150 mM NaCl, 0.4 mg mL$^{-1}$ BSA, 0.005% P20, 2% DMSO). All compounds were prepared in 100% DMSO stock plates with a top concentration of 500 μM in a 3× serial dilution. Compounds were transferred from the stock plate to the assay plate and diluted into running buffer without DMSO. Equimolar concentration of target protein was added to corresponding compound wells. All compounds were run as a 12-point concentration series with a final assay top concentration of 50 μM to measure p38δ:PROTAC and

VHL:PROTAC binding and top concentration of 5 μM to measure p38δ:PROTAC:VHL binding. All compounds were injected for 60 s and allowed to dissociate for 300 s. Data analysis was performed in Scrubber 2 (BioLogic Software). Blanks from reference flow cell containing immobilized GST-VBC were subtracted to correct for noise and data were solvent-corrected against a standard DMSO curve. All reported $K_d$ values represent an average of at least two replicates and were obtained by fitting to a minimum of five concentrations using a 1:1 fitting algorithm.

**Ubiquitination assays**. HeLa cells ($2 \times 10^6$) were seeded into 10 cm dishes and allowed to adhere overnight. On the following day, cells were transfected with 3 μg pcDNA5-FLAG-p38α or pcDNA5-FLAG-p38δ and 1 μg pRK5-HA-Ubiquitin-WT (Addgene plasmid #17608) in Opti-MEM media using Lipofectamine 2000 (Invitrogen). After 6 h, Opti-MEM media were replaced with DMEM ($1 \times$) and cells were grown for an additional 24 h. After this time, cells were treated with indicated concentrations of either vehicle (DMSO) or PROTAC for the indicated amount of time at 37 °C. Cells were then placed on ice, rinsed twice with ice-cold 1X PBS and lysed in 500 μL modified 1X RIPA buffer (25 mM Tris-HCl pH 7.6, 150 mM NaCl, 1% NP-40, 1% sodium deoxycholate, 0.1% SDS) containing 5 mM 1,10-phenan-throline monohydrate, 10 mM N-ethylmaleimide, 20 μM PR-619, and 1X protease inhibitor cocktail (Roche). Lysates were spun down at $14,000 \times g$ at 4 °C for 20 min and protein content was measured by Pierce BCA Protein Assay (ThermoFisher Scientific). Protein lysate was normalized and 1 mg of lysate was aliquoted onto 20 μL (bed volume) of DYKDDDDK-sepharose beads (CST: #70659). FLAG-containing proteins were immunoprecipitated from HeLa lysates overnight at 4 °C with gentle rotation, after which samples were spun down at $3000 \times g$ at 4 °C for 2 min and the beads were washed once with ice-cold lysis buffer and three times with ice-cold 1X TBS-T (137 mM NaCl, 2.7 mM KCl, 19 mM Tris-HCl pH 7.5, 0.02% Tween 20). Beads were resuspended in 1X lithium dodecyl sulfate (LDS) sample buffer containing 5% 2-mercaptoethanol (BME). Immunoprecipitated protein was eluted off of the beads by heating at 95 °C for 5 min and the super-natant was run on an SDS-PAGE gel and evaluated for the presence of immu-noprecipitated FLAG-tagged proteins (anti-FLAG M2, Sigma #F1804), as well as ubiquitinated FLAG-tagged proteins (anti-HA-tag (6E2), CST #2367). WCL refers to the normalized input lysate loaded onto DYKDDDDK-sepharose beads.

TUBE1 immunoprecipitation experiments were carried out exactly as described above, except for the fact that 1 mg of normalized HeLa lysate was loaded onto 20 μL TUBE1 agarose (LifeSensors) resin per sample.

**Quantitative real-time PCR**. MDA-MB-231 cells were plated at $3 \times 10^5$ cells per well in a six-well dish, allowed to adhere, and were treated with either vehicle (DMSO) or PROTAC (250 nM) for 24 h. Cell were lysed in 1 mL of TRIzol reagent (ThermoFisher Scientific) per six well. Chloroform was added (200 μL) per sample, after which samples were vortexed vigorously for 15 s, and centrifuged at $12,000 \times g$ for 15 min at 4 °C. Total RNA was then precipitated from the aqueous phase by addition of isopropanol (500 μL), followed by centrifugation at $12,000 \times g$ for 10 min at 4 °C. RNA pellet was washed twice with 75% EtOH and allowed to air dry for 15 min, after which RNA was dissolved in 25 μL of nuclease-free H$_2$O. Complementary DNA was synthesized from 4 μg of total RNA per condition according to the manufacturer's protocol (Applied Biosystems) and real-time PCR was performed with 800 nM primers, diluted with 4 μL SYBR Green Reaction Mix (Applied Biosystems). RT-PCR experiments were performed with the following protocol on a LightCycler 480 Instrument II (Roche): 95 °C for 10 min, 40 cycles of 95 °C for 15 s, and 60 °C for 45 s. qRT-PCR samples were performed and analyzed in triplicate, from two independent experiments. Beta-tubulin was used for nor-malization. Primers used in this study are as follows:

Beta_Tub_F: 5′-TGGACTCTGTTCGCTCAGGT-3′
Beta_Tub_R: 5′-TGCCTCCTTCCGTACCACAT-3′
p38α_F: 5′-TCGCATGAATGATGGACTGAAAT-3′
p38α_R: 5′-CCCGAGCGTTACCAGAACC-3′
p38δ_F: 5′-TGAGCCGACCCTTTCAGTC-3′
p38δ_R: 5′-AGCCCAATGACGTTCTCATGC-3′.

**CHX chase assay**. MDA-MB-231 cells were plated at $3 \times 10^5$ cells per well in a six-well dish, allowed to adhere overnight, and switched to serum-free RPMI-1640 ($1 \times$) media for 16 h. Cells were then pre-treated with CHX (Sigma) at 100 μg mL$^{-1}$ for 1 h prior to adding either vehicle (DMSO) or PROTAC (250 nM). At the indicated timepoints, cells were immediately placed on ice, rinsed with 1X PBS, lysed, and boiled.

**PROTAC washout assay**. MDA-MB-231 cells were plated at $1.5 \times 10^5$ cells per well in a six-well dish and allowed to adhere overnight. On the following day, all cells were treated with either vehicle (DMSO) or PROTAC (250 nM). After 24 h, cells were either harvested or washed with 1X PBS, trypsinized, and re-plated onto new six-well dishes in fresh RPMI-1640 ($1 \times$) media without additional compound. For these washout conditions, cells were collected every 24 h until final harvest (72 h washout).

**MD simulations**. The starting coordinates for p38δ came from the crystal structure downloaded from Protein Data Bank (PDB) entry . In order to replace the ligand in

this structure with foretinib, the PDB entry 5IA4 was used by overlaying its protein backbone with that of 4EYJ, transferring foretinib to the 4EYJ structure and replacing the original ligand. The obtained p38δ–foretinib complex was subject to the Protein Preparation Wizard of Maestro 2016-3 program available from Schrodinger Inc. (New York City, NY), with which the hydrogen atoms were added, the missing side chains were built, and the protonation states were assigned assuming a pH of 7.0 for the ionizable groups. An energy minimization of the complex was performed for 500 steps.

The starting coordinates for VHL came from the PDB entry 4W9H. The starting coordinates for the p38δ:PROTAC:VHL trimer were prepared as follows. (1) The electrostatic surface was generated for p38δ–ligand complex and VHL ligand complex, respectively. (2) The VHL ligand complex was set to have different relative dispositions with respect to the p38δ–ligand complex in a way that the hydrophobic patch of the VHL ligand surface opposed different hydrophobic patches and grooves of the p38δ–ligand surface, thus producing different starting modes in terms of the relative dispositions between p38δ and VHL. (3) For each starting mode, a linker was built to connect foretinib and VHL ligand and form the full PROTAC. And (4) an energy minimization of 500 steps was performed for each starting point of trimer.

OPLS3 force-field was used throughout the calculation steps. The torsional angle parameters were examined with Force Field Builder program and found that the torsional angles between the amide and cyclopropyl group and between the fluorophenyl group and the oxygen ether atom attached to the quinoline group in foretinib needed corrections; and thus the new torsional profiles were generated to match the profiles given by Jaguar quantum mechanical calculations.

Each starting point of the p38δ:PROTAC:VHL trimer was subject to MD simulation. The system setup was done using System Builder of Maestro program, in which the periodic boundary condition was used; the box shape was cubic with absolute size of each side greater than the largest dimension of the system by 5 Å. The explicit waters were added. The system was neutralized using sodium and chloride ions and salted into 0.15 M ionic strength. The MD was done using Desmond Multisim version 3.8.5.19, which has an eight-stage process: (1) task; (2) simulation of 100 picosecond with Brownian dynamics NVT with T at 10 K, small time-steps and restraints on solute heavy atoms; (3) simulation of 12 picosecond, NVT ensemble, T at 10 K, small time-steps and restraints on solute heavy atoms; (4) simulation of 12 picosecond, NPT ensemble, T at 10 K and restraints on solute heavy atoms; (5) solvation of unfilled pockets; (6) simulation of 12 picosecond up to the target temperature of 310 K, NPT ensemble and restraints on solute heavy atoms; (7) simulation of 24 picosecond, NPT ensemble without restraints at T of 310 K; and finally, (8) production run of 100 nanoseconds. During the production run, coordinate frames were saved at every 10 picoseconds. The target pressure was set to 1.01325 bar in the related steps.

The post-simulation analysis after each run was done as follows. The last 20 nanoseconds of trajectory frames were extracted. A clustering analysis using hierarchical clustering method was performed. The distance between any two members (frames) was the root-mean-square deviation of the solute heavy atoms between the members after overlaying them. The cutoff distance was 2 Å. Every frame was used. The structure closest to the centroid of each cluster was written out as the representative structure of that cluster. The representative structure of the largest cluster for each MD simulation was considered as the representative structure of that simulation run. Such a structure can be considered as the most-visited conformation of that run.

The MD simulation was performed using the g2.2 × large instances of Amazon Web Service cloud machines. The Desmond GPU-enabled code was used and mainly run using GPU.

**Chemical syntheses**. Details of PROTAC chemical syntheses can be found in the Supplementary Information as Supplementary Note 1.

**Quantification and statistical analysis**. Western blot data in Figs. 2, 6, and Supplementary Figures 2, 3 were quantified by using the band feature in Image Lab (Bio-Rad). The p38 values were normalized to the α-tubulin signal to account for differences in protein load, and PROTAC-treated normalized values were expressed as a % of the vehicle-treated (no PROTAC) value, which itself was defined as 100% p38 protein level. These data were then graphed and fit to a sigmoidal curve by non-linear regression using Prism data-fitting software (Graphpad Prism). The sigmoidal fit established both the full extent of degradation, or $D_{max}$ at the bottom asymptote of the curve, as well as the $DC_{50}$, which is the inflection point at which 50% of the total degradation observed was reached.

**Reporting summary**. Further information on experimental design is available in the Nature Research Reporting Summary linked to this article.

## Data availability

A reporting summary for this article is available as a Supplementary Information file. All data supporting the findings of this study are available from the corresponding author upon reasonable request.

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

## Acknowledgements

We thank all members of the C.M.C. laboratory for helpful discussion throughout experimentation, John Hines for manuscript edits, and Katie Roberts for technical support. C.M.C. gratefully acknowledges support from the NIH (R35CA197589) and Arvinas, Inc.

## Author contributions

B.E.S., S.J.-F., S.L.W., and C.M.C. conceived the project. B.E.S., S.L.W., S.J.-F., A.H., J.W., and B.D.H. planned and carried out experiments. B.E.S. wrote the manuscript. All authors reviewed and edited the manuscript.

## Additional information

**Competing interests:** C.M.C. is a consultant and shareholder in Arvinas, Inc. A.H., J.W., and B.D.H. are employees and shareholders of Arvinas, Inc. The remaining authors declare no competing interests.

