## [Peer Review File · Nature Communications]

Reviewers' comments:

Reviewer #1 (Remarks to the Author):

This manuscript describes the development of selective PROTACs targeting p38 isoforms by varying the linker connecting the warhead and E3 ligase ligand. This is an exceptionally elegant demonstration of the importance of the linker region for PROTAC efficacy. p38alpha is a target for inflammation, though as yet p38 inhibitors have not been successful in the clinic, while little is known about the physiological role of p38delta, so the potential of these reagents in drug development and as functional tools adds significance to this work. The authors should address the following issues:

Major

1. It is unclear how the authors arrived at the values of DC50 and Dmax, especially with respect to SJFdelta. The authors report only a lower limit for DC50 for SJFdelta. However, in the usual data analysis, Dmax would be the maximal response, and DC50 would be the concentration that gives half the maximal response. There can't be a value for Dmax without at least an upper limit for the value for DC50. Therefore, for SJFdelta, DC50 would be the concentration that causes 17% degradation. Also, errors should be reported in the Table 1 and Figure 1 (not hidden in the supporting information). The dose response curves and corresponding fits should be provided in the supporting information.
2. Figure 1 does not include sufficient data to assess the validity of values of DC50 and Dmax. For example, the lowest concentration of SJFalpha is 25 nM, but the DC50 is reported to be 9.5 nM. Presumably the authors have more complete dose response data- this should be included in the figure. The figure should also include quantitation of the gel.
3. Figure 3, panel C is not very convincing: the authors immunoprecipitate FLAG-p38alpha, and then probe HA-Ub and see a smear of the of high MW ubiquitin. They conclude that p38alpha is ubiquitinated. However, only one band of FLAG-p38alpha is shown, suggesting that the high MW ubiquitin species may be proteins that co-immunoprecipitated with p38alpha. The same issue applies to Figure 4C. The TUBE1 experiment in the supporting figure S6 is a much more convincing.

4. It is unclear how the authors are calculating residence time in Table 1. Residence time is simply $1/k_{off}$ (see Copeland NRDD 2007), which would be 57 s for SJFdelta not 330 s as reported in Table 1. In any case, residence time is redundant with half-life- only one value should be reported.

5. The potential of a hook effect complicates the interpretation of the CESTA experiments. No hook effect was observed at PROTAC concentrations of up to 5 μ M. However, the CETSA experiments were performed at 100 μ M PROTAC. The authors must demonstrate the absence of a hook effect at this concentration to ensure that the CETSA experiment does not report on the stability of the binary complex rather than the ternary complex.

6. Discussion lines 474-476 are confusing... if a concentration is above the K_d , then there should be a significant accumulation of ternary complex. However, SJFalpha concentration of 5 μ M refers to the concentration outside the cell- the concentration inside the cell is unknown, and presumably below the K_d . Note that the concentrations of the proteins will also be important. This should be clarified.

Minor Issues

1. It is unlikely that these experiments can be performed with a precision of more than 2 significant figures. Too many significant figures are an issue throughout the paper- especially for Table 1 (5 significant figures in some entries!).

2. The authors should consider describing the results with the non-selective PROTACs first, then focus on further characterization of the selective PROTACs- this could improve the flow of the manuscript.

3. Supporting figures: the authors use the labels N=1 etc when they probably mean Replicate #1.

4. Figure S4B should be scatter plots so that the values can be more easily compared at each temperature. Note that there are no labels for the actual temperatures on the x-axis.

5. Figure S5 C- the fits to the SPR data should be included. The panels should be larger and higher resolution (it is impossible to read the key).

6. line 317 typo- koff for SJFalpha is 0.072 s⁻¹ according to Table 1.

Reviewer #2 (Remarks to the Author):

The article describes isoform selective PROTACs for the p38 MAPK family. The authors explored linker lengths and attachment points on the PROTAC dimers consisting of ligands for p38 and E3. A series of glycol derivatives were tested as linkers. The goal, beyond creating selective ligands for p38, is to develop general design principles to guide selectivity of degraders. Both questions are important: development of selective p38 degraders is an important medicinal chemistry question but the second question is critical for further development of PROTACs. Thus the study is well suited for Nature Comm.

The paper succeeds in developing potent and selective ligands for p38 α and p38 δ respectively without affecting p38 β and p38 γ isoforms. This is remarkable; that differences in orientation can drastically alter degradation. However, the paper raises more questions than answers on the second, biophysical, question. Since the focus of the paper is largely on the biophysical aspects, it is difficult to recommend its publication until key experimental results are better addressed.

Through pull-down experiments the authors show that enrichment of p38 α occurs in vitro with one derivative (SJF α), and that the resulting poly-ubiquitinylation in vivo is responsible for target degradation. In contrast SJF δ mediated degradation is a direct result of subtle differences in ternary complex affinity; the keen observation that both SJF α and SJF δ can bind the same target (p38 δ) but that only SJF δ binding results in degradation is novel. With subsequent SPR, CETSA and ubiquitinylation assays showing that cellular engagement between SJF δ and p38 δ is more prolific than that of SJF α and p38 δ , Smith et al. suggest that increased association of SJF δ and p38 δ with VHL is enough to result in profoundly different cellular ubiquitination.

The take home message, then is: the linker is not innocuous but should be treated more formally as part of the ligand. This is not a surprising conclusion but some of the results are difficult to comprehend. The paper suggests that single atom alterations to a flexible PEG linker (10 to 14 atoms in length) can fully alter the degradation profiles. Table S1 lists several derivatives and their degradation potencies. The differences in results for SJF6690 versus SJF-8240 and SJF δ versus SJF-6677 for p38 δ are huge for a single atom in a flexible linker. The authors focus the paper on the

profiles of SJF α and SJF δ but largely ignore the single-atom differences leading to profound reactivity differences.

Several controls and other biophysical assays that would get to the important biophysical questions addressing the role of specificity should be performed.

— The Brd4:PROTAC:VHL complex (PDB 5T35) suggests that the "linker" is not just an innocuous connection between the 2 halves of the molecule. The glycol units linking the two ligands likely make its own contacts. The question then is at what point does it just become part of the ligand? It would be nice to see data with each ligand and linker fragments to see if the linker is directly involved in binding of either protein. Also, what parts of the molecule are necessary for ternary complex formation - are both ligands required? One would assume so, but it's not explicitly demonstrated. Otherwise, the linker just controls the relative ligand orientation and follows the more traditional philosophy of a linker.

— The MD simulations are a starting point and show different orientations of VHL:PROTAC:p38 δ complexes with different PROTACS given that their linkers are fairly flexible. The docked structures are possible but cross-linking studies (or maybe mutagenesis) would provide better evidence for the notion that each PROTAC can promote different protein-protein contacts. On this front, the paper is weak in terms of literature context. There are several instances where a molecule, in binding to a protein, generates a hybrid protein-small molecule surface that is a better binding site for another protein. This is the mode of action for molecules like cyclosporin and rapamycin, where non-native protein-protein contacts are too limited to be effective in the absence of the small molecule but enhances complex formation when the small molecule is present.

Overall, the initial findings of this paper are confusing; that single atom linker length and orientation differences between warhead and recruited E3 ligase can provide orthogonal degradation of two closely related protein isoforms. The med chem is good and the targets are appealing. But what is the bigger picture insight? That PROTACs are more subtle than originally thought? The same can be said for any ligand design process.

Reviewer #3 (Remarks to the Author):

The manuscript entitled "Differential PROTAC Substrate Specificity Dictated by Orientation of Recruited E3 Ligase" from the Crews lab describes the synthesis and evaluation of new PROTAC

ligands derived from the promiscuous kinase inhibitor foretinib. This work builds on their recent publication called "Lessons in PROTAC design from selective degradation with a promiscuous warhead". In this submission the team describes two related PROTAC reagents that are able to differentially degrade two closely related kinases. One of the ligands, SJF α , leads to degradation of P38 alpha (gene name MAPK14). The other, SJF δ , leads to degradation of P38 delta (gene name MAPK13). My recommendation is that this paper be published with minor corrections.

The major finding here is that subtle changes of the PROTAC reagents have different biophysical consequences, and thus can result in different pharmacology. This in and of itself is not surprising conclusion as practitioners of medicinal chemistry will well understand. PROTAC ligands have increased complexity compared to single target small molecule medicinal chemistry efforts. Structure activity relationships exist for the warhead (foretinib here), the E3 ligase recruiting moiety, and the linker. These structure activity relationships are all important and are not independent. The Crews team has done a great job describing an example of exploring two out of three of these structure activity relationships together and demonstrate that the careful medicinal chemistry allow one to identify tools with different and useful pharmacology. I think this manuscript represents a valuable addition to the field.

My experience is that people now think that if you have a ligand you can hook on a linker and hook on an E3 ligase recruiting molecule and knock down your target. A paper like this highlights the promise of this strategy, but also provides a sobering dose of realism. There are things we do not understand, and you need to explore many structural aspects of these PROTAC ligands in order to optimize for the desired biological response. The demonstration that modifications of linker length and orientation of the VHL-recruiting molecule in concert can allow for the degradation of P38 alpha or P38 delta is an exciting finding that will influence the way people approach PROTAC projects. Demonstrating different pharmacology for the so-called amide series and phenyl series is important and should make people aware that there is a very broad range of permutations that need to be explored to optimize these tools, and ideally convert them into medicines.

The authors have done a good job demonstrating that structure modulation modulates pharmacology. They have used appropriate control experiments to show that the degradation is occurring via the expected mechanism. An additional finding that is important to the field is that degradation is related to the stability of the ternary complex. Other papers have demonstrated that warhead affinity by itself does not mean knockdown is inevitable. I think this paper will lead to more structural characterizations of these complexes and new assays to effectively measure the parameters that will help us understand knockdown and screen for compounds that will be effective.

I have a few suggestions of things I would change.

I don't think P38 delta should be called undruggable. It certainly has not been targeted as much as P38 alpha, but one can find inhibitors (not necessarily selective) in the primary literature (Roche compound 17 in <https://www.ncbi.nlm.nih.gov/pubmed/23664880>; <https://www.ncbi.nlm.nih.gov/pubmed/27369736>) or the recent patent literature. I'm convinced that if a team wanted to make selective P38 delta inhibitors they could. To me a conclusion of this manuscript is that detailed SAR exploration and appropriate assays can allow one to identify selective degraders of closely related targets – the undruggable part does not follow from this specific example necessarily.

I also wish the authors would reiterate that foretinib is a promiscuous kinase inhibitor and engages a broad range of kinases. If people have read the previous papers and are very familiar with kinase inhibitors they will know this, but this detail it is not explicit enough in this manuscript. So although selectivity between PI3K alpha and PI3K delta has been obtained and this is an exciting and important achievement, broad kinome knockdown selectivity has not necessarily been achieved with these two molecules because those experimental results have not been described here. I suggest that wording about selectivity be carefully chosen. If in fact you know other proteins are degraded outside of P38 alpha and P38 delta, then those results should be shared. If these data don't exist, then the conclusion is that exquisite selectivity between alpha and delta has been achieved, but broader effects are yet to / need to be elucidated.

Finally some typos:

On page 5 thienopyrinone should be thienopyridinone (line 122)

In the experimental:

Scheme 1:

Acetona should be acetone

PRPTAC SJF-6690 should be PROTAC SJF-6690

Thank you for this contribution. I thoroughly enjoyed reading it and revisiting in greater detail some of the references you referred to. I am confident you have provided enough detail (from this and related manuscripts) that your work can be repeated and that this will influence the field.

Reviewer #1

Major concerns

“It is unclear how the authors arrived at the values of DC_{50} and D_{max} , especially with respect to SJFdelta. The authors report only a lower limit for DC_{50} for SJFdelta. However, in the usual data analysis, D_{max} would be the maximal response, and DC_{50} would be the concentration that gives half the maximal response. There can't be a value for D_{max} without at least an upper limit for the value for DC_{50} . Therefore, for SJFdelta, DC_{50} would be the concentration that causes 17% degradation.

Also, errors should be reported in the Table 1 and Figure 1 (not hidden in the supporting information). The dose response curves and corresponding fits should be provided in the supporting information.”

In the revised version of our manuscript, we have

In order to determine the DC_{50} and D_{max} values, the western blot band intensities for p38 and tubulin were each measured using a ChemiDoc Gel Imaging System (Bio-Rad) designed to directly measure chemiluminescent signal. The p38 values were normalized to the tubulin signal to account for differences in protein load, and PROTAC treated normalized values were expressed as a % of the vehicle-treated (no PROTAC) value, which itself was defined as “100%” p38 protein level (which we assume is the upper limit of which Reviewer 1 writes above). These data were then fit to a sigmoidal curve using data fitting software (GraphPad Prism), which established the full extent of degradation, or D_{max} at the bottom asymptote of the curve, as well as the DC_{50} , which is the inflection point at which 50% of the total degradation observed was reached. A more explicit depiction of the degradation of the p38 isoforms was analyzed this way and has now been included in the revised manuscript (Fig. S3, Supplemental Table 1). We have also added a more explicit accounting of our analysis method to the Methods section so that there will now be no ambiguity to the reader. Error estimations of our data are also now included in the manuscript proper.

“Figure 1 does not include sufficient data to assess the validity of values of DC_{50} and D_{max} . For example, the lowest concentration of SJFalpha is 25 nM, but the DC_{50} is reported to be 9.5 nM. Presumably the authors have more complete dose response data- this should be included in the figure. The figure should also include quantitation of the gel.”

The Reviewer makes a valid point here and we have now included more extensive dose-response western blots for both SJF-alpha and SJF-delta in the revised manuscript (Fig. S3) that includes concentrations below the DC_{50} value. Furthermore,

sigmoidal plots of the degraded proteins are also included so that the veracity of the stated D_{max} and DC_{50} values will be apparent.

“Figure 3, panel C is not very convincing: the authors immunoprecipitate FLAG-p38alpha, and then probe HA-Ub and see a smear of the of high MW ubiquitin. They conclude that p38alpha is ubiquitinated. However, only one band of FLAG-p38alpha is shown, suggesting that the high MW ubiquitin species may be proteins that co-immunoprecipitated with p38alpha. The same issue applies to Figure 4C. The TUBE1 experiment in the supporting figure S6 is a much more convincing.”

We understand Reviewer 1’s point about the lack of higher MW conjugates in the western blots of the FLAG-tagged p38 constructs (Figure 3C and 4C) in the ubiquitination assays. Because the anti-ubiquitin (anti-HA) and anti-FLAG blots are both probing p38 in the same lysates in these figures, it is understandable that *a priori* one might expect to see an identical pattern and then wonder why ubiquitinated conjugates are detected only in the anti-ubiquitination blots. This is, in fact, commonplace in experiments such as these that rely on overexpression of the ubiquitination target (for ex., see Krönke *et al.* (2014) *Science* **343**: 301-305; Dai *et al.* (2017) *Nat. Med.* **23**: 1063-1071). The apparent disconnect stems from the overexpression levels of the exogenous substrate – in this case FLAG-tagged p38 isoforms – being much greater than the capacity of the UPS to accommodate in response to the PROTAC treatment interval. Thus, while some percentage of the FLAG-tagged p38 gets ubiquitinated in response to the PROTAC treatment, there is an even larger pool of the overexpressed p38 that is beyond what the UPS can process and therefore fails to be ubiquitinated. Since the anti-FLAG pulldown recognizes *all* the overexpressed FLAG-p38 -- much of which is not ubiquitinated -- the FLAG blot of the pulldown reflects this. Because the anti-HA blots are specific for that % of the FLAG-p38 pulldown that had been ubiquitinated, the higher MW conjugates appear in that blot only. Unfortunately, without the overexpression of the p38 isoforms, we are unable to detect PROTAC-dependent ubiquitination: reliance on only the endogenous p38 to see higher MW ubiquitin conjugates did not yield a sufficiently strong signal, even in the presence of epoxomicin to inhibit the proteasome.

“It is unclear how the authors are calculating residence time in Table 1. Residence time is simply $1/k_{off}$ (see Copeland NRDD 2007), which would be 57 s for SJFdelta not 330 s as reported in Table 1. In any case, residence time is redundant with half-life - only one value should be reported. “

We thank the Reviewer for this comment and we apologize for the confusion. In the original manuscript, we were referring to the residence time of the ternary complex, which we defined as the amount of time after injection stops ($t=60$ s) for the ternary complex to completely dissociate (i.e. reach 0 RU). To avoid any confusion, we have chosen to remove this residence time of the ternary complex and in our revised

manuscript we instead use half-life as a representative measurement of the ternary complex and have reflected these changes in the text and the Methods section.

“The potential of a hook effect complicates the interpretation of the CESTA experiments. No hook effect was observed at PROTAC concentrations of up to 5 μ M. However, the CETSA experiments were performed at 100 μ M PROTAC. The authors must demonstrate the absence of a hook effect at this concentration to ensure that the CETSA experiment does not report on the stability of the binary complex rather than the ternary complex.”

Thank you to the Reviewer for another fine point. To address this, we extended our dose-response that we performed to address Reviewer 1’s second point (above) to include PROTAC concentrations up to 100 μ M (see Fig. S3). As shown, there was no significant hook effect observed at these concentrations.

“Discussion lines 474-476 are confusing... if a concentration is above the K_d , then there should be a significant accumulation of ternary complex. However, SJFalpha concentration of 5 μ M refers to the concentration outside the cell - the concentration inside the cell is unknown, and presumably below the K_d . Note that the concentrations of the proteins will also be important. This should be clarified.”

Thank you for this point. We have removed this sentence from the Discussion to avoid any confusion or convolution regarding cell-free and in-cell concentrations.

Minor concerns

“It is unlikely that these experiments can be performed with a precision of more than 2 significant figures. Too many significant figures are an issue throughout the paper- especially for Table 1 (5 significant figures in some entries!).”

Thank you for this point. We have now fixed this in the revised manuscript.

“The authors should consider describing the results with the non-selective PROTACs first, then focus on further characterization of the selective PROTACs- this could improve the flow of the manuscript.”

The first section of the Results have been revised as per Reviewer 1’s suggestion here.

“Supporting figures: the authors use the labels N=1 etc when they probably mean Replicate #1.”

This has now been revised as per Reviewer 1’s suggestion.

“Figure S4B should be scatter plots so that the values can be more easily compared at each temperature. Note that there are no labels for the actual temperatures on the x-axis.”

This has now been revised as per Reviewer 1’s suggestion.

“Figure S5 C- the fits to the SPR data should be included. The panels should be larger and higher resolution (it is impossible to read the key).”

This has now been revised as per Reviewer 1’s suggestion.

“Line 317 typo- koff for SJFalpha is 0.072 s-1 according to Table 1.”

This has now been revised as per Reviewer 1’s suggestion.

Reviewer #2

Major concerns

“The take home message, then is: the linker is not innocuous but should be treated more formally as part of the ligand. This is not a surprising conclusion but some of the results are difficult to comprehend. The paper suggests that single atom alterations to a flexible PEG linker (10 to 14 atoms in length) can fully alter the degradation profiles. Table S1 lists several derivatives and their degradation potencies. The differences in results for SJF6690 versus SJF-8240 and SJF δ versus SJF-6677 for p38 δ are huge for a single atom in a flexible linker. The authors focus the paper on the profiles of SJFa and SJF δ but largely ignore the single-atom differences leading to profound reactivity differences.”

We thank Reviewer 2 for the attention that they have given to our study – however, we do not believe that the point raised by Reviewer 2 has been ignored: the functional effects of the single atom differences in the linkers are a clear and present conclusion that can be drawn from our study, and one which has been acknowledged and addressed in our Discussion section. However, this study has focused on the profound

outcomes of ternary complex formation with distinct kinase family members that proceed from the simple structural changes that were implemented in our PROTAC; and that much smaller structural changes than those implemented in previous published studies can alter PROTAC substrate specificity in ways that, certainly in the case of the p38 kinase family, inhibitors have not yet achieved. The take home point of the study goes beyond the ability of an extra methylene group to alter the extent of engagement of a single target protein; that is a topic that we have, actually, covered in detail in a previous publication [Buckley *et al.* (2015) *ACS Chem. Biol.* **10**(8): 1831-1837]. As we mention throughout the paper, subtle design principles – from single atomic linker changes to the orientation of the recruited E3 ligase (amide vs. phenyl series – have drastically changed both the cellular potency, but also the selectivity of each PROTAC (and each PROTAC series/class) for the p38 isoforms. As this Reviewer mentions, we then go on to acquiesce that the linker is not just an innocuous component of a given PROTAC. Thus, the current study attempted to shed light on how these subtle changes can be co-opted for future PROTAC design. In the Results and Discussion, we emphasize the role that future linker exploration and E3 ligase recruitment geometry can play in producing selective degradation outcomes. That is, we hope that this work will spur future PROTAC work, with an emphasis on further exploring linker space, linker composition, E3 ligase recruitment geometry, and E3 ligase choice as part of the PROTAC ‘toolkit’ going forward.

“The Brd4:PROTAC:VHL complex (PDB 5T35) suggests that the “linker” is not just an innocuous connection between the 2 halves of the molecule. The glycol units linking the two ligands likely make its own contacts. The question then is at what point does it just become part of the ligand? It would be nice to see data with each ligand and linker fragments to see if the linker is directly involved in binding of either protein. Also, what parts of the molecule are necessary for ternary complex formation - are both ligands required? One would assume so, but it's not explicitly demonstrated. Otherwise, the linker just controls the relative ligand orientation and follows the more traditional philosophy of a linker.”

There is an evolving recognition in the PROTAC field that the mechanistic involvement of the linker in mediating dimerization of the target protein with the E3 ligase can be more active than previously assumed, and we have reviewed how our work reflects this in the second and third paragraphs of our Discussion. However, this point was not a main thesis of the paper, but rather one of our closing points. Certainly, Reviewer 2’s suggestions here are interesting and could form the foundation for a subsequent and more mechanistic study on isoform-discriminating PROTACs; however it was not our intention to explore those fine mechanistic details of the ‘linkerology’ in the present study, but rather to demonstrate the practical application of the largely untapped and therefore underappreciated potential for fine tuning of substrate specificity through relatively subtle variations in PROTAC structure.

The Reviewer’s implication that the removal of either of the tethered ligands could have nominal impact on PROTAC-induced target degradation seems unlikely. In several

instances, work by our group and others have shown that the mere switching of a key stereocenter in the VHL ligand abolishes degradation activity despite the continued inclusion of the same linker in the epimeric PROTAC variant. That so many PROTACs (for HER1, HER2, c-Met, ERRalpha, RIPK2, etc.) can induce productive PPIs between their targets and VHL, but then are completely unable to mediate degradation when a single stereocenter is inverted demonstrates, in our opinion, a critical reliance on the unadulterated presence of both ligands for achieving the effect. Therefore, we strongly believe that in the absence of either ligand (i.e. warhead or E3 recruiting ligand), even with a linker attached, PPIs would not form. However, aside from demonstrating how the linker region can “tune” the affinity contributions of the ligands -- through steric limitations/allowances or through affinity contributions of its own -- detailed or comprehensive structural considerations of linker interaction with the binding proteins are beyond the scope of this study.

“The MD simulations are a starting point and show different orientations of VHL:PROTAC:p38delta complexes with different PROTACs given that their linkers are fairly flexible. The docked structures are possible but cross-linking studies (or maybe mutagenesis) would provide better evidence for the notion that each PROTAC can promote different protein-protein contacts. On this front, the paper is weak in terms of literature context. There are several instances where a molecule, in binding to a protein, generates a hybrid protein-small molecule surface that is a better binding site for another protein. This is the mode of action for molecules like cyclosporin and rapamycin, where non-native protein-protein contacts are too limited to be effective in the absence of the small molecule but enhances complex formation when the small molecule is present.”

We thank Reviewer 2 for this point. In the way that it has been presented, we believe that these models have offered a glimpse into *possible* ternary complex binding modes. These models were meant to offer a launch point for future structural work (by our lab or by others). To support our model of the PROTAC-induced molecular interactions during complex formation, we compared the MD simulation of p38delta and VHL complexed in the presence of the degrader SJFdelta to the predicted protein-protein interactions that would occur between these same proteins in the presence of SJFalpha (the non-degrading PROTAC for p38delta). Our model predicted that a pair of amino acids in p38delta – Lys220 and Thr221 – would make unfavorable interactions with a key Arg in VHL (Arg108) in the presence of SJFalpha; whereas in the presence of SJFdelta, a different pair of amino acids – Glu49 and Glu160 – make stabilizing electrostatic interactions. To verify this, we performed mutagenesis, as per Reviewer 2’s suggestion, to see whether substitution of Lys220 and Thr221 with Glu residues might permit for a more favorable interaction with Arg108 of VHL in the presence of SJFalpha (non-degrader) in a similar manner to that of VHL Arg108 and p38delta Glu49 and Glu160 (when in the presence of the degrader SJFdelta) . We found that, when expressed in cells and tested, SJFalpha is now able to cause degrade the double-mutant p38delta and SJFdelta is able to degrade this double-mutant p38delta with equipotency to WT p38delta, demonstrating the validity of both p38delta ternary complex models. This data

has been included in the revised manuscript, with the entirety of our presentation of MD simulations having now been relocated to the Results section.

“Overall, the initial findings of this paper are confusing; that single atom linker length and orientation differences between warhead and recruited E3 ligase can provide orthogonal degradation of two closely related protein isoforms. The med chem is good and the targets are appealing. But what is the bigger picture insight? That PROTACs are more subtle than originally thought? The same can be said for any ligand design process.”

Given this point, this Reviewer does not appear to have been confused by the study; rather, he/she appears to have understood the major findings of the paper. Our stated goal in this work was to develop degraders that are selective for either of two closely-related isoforms of a protein family. Unlike in our previous studies, which modulated target specificity by wholesale exchanging of targeting warheads or E3 ligase ligands, in the current study we leaned on the role of linker length and the underexplored role of recruited E3 ligase geometry to generate specificity. In doing so, we were able to achieve highly discriminating degradation behavior *without* changing the targeting warhead or the E3 ligase recruited. Moreover, one of the targets we successfully degraded, p38delta, is an isoform that has been especially difficult to target using the occupancy-based strategy of inhibitors.

Reviewer #3

Major concerns

“I don't think p38 delta should be called undruggable. It certainly has not been targeted as much as p38 alpha, but one can find inhibitors (not necessarily selective) in the primary literature (Roche compound 17 in <https://www.ncbi.nlm.nih.gov/pubmed/23664880>; <https://www.ncbi.nlm.nih.gov/pubmed/27369736>) or the recent patent literature. I'm convinced that if a team wanted to make selective p38 delta inhibitors they could. To me a conclusion of this manuscript is that detailed SAR exploration and appropriate assays can allow one to identify selective degraders of closely related targets – the undruggable part does not follow from this specific example necessarily.”

The reviewer makes a very good point regarding the important difference between the 'undruggable' proteome (proteins with no tractable active sites) versus 'druggable' kinases with active sites that are especially challenging to exploit using inhibitors. The former have not been engaged due to lack of viable strategy, while attempts have been made at engaging the latter but have not yet resulted in the identification of a lead compound. The p38delta MAPK falls into the latter category, and accordingly we have removed the descriptions of that isoform as “undruggable” in favor of describing it as

“refractory to inhibition” (see the Significance Statement; Introduction 5th paragraph; Discussion final paragraph).

“I also wish the authors would reiterate that foretinib is a promiscuous kinase inhibitor and engages a broad range of kinases. If people have read the previous papers and are very familiar with kinase inhibitors they will know this, but this detail it is not explicit enough in this manuscript. So although selectivity between PI3K alpha and PI3K delta has been obtained and this is an exciting and important achievement, broad kinome knockdown selectivity has not necessarily been achieved with these two molecules because those experimental results have not been described here. I suggest that wording about selectivity be carefully chosen. If in fact you know other proteins are degraded outside of P38 alpha and P38 delta, then those results should be shared. If these data don't exist, then the conclusion is that exquisite selectivity between alpha and delta has been achieved, but broader effects are yet to / need to be elucidated.

The point Reviewer 3 raises here is a fine one, indeed. The distinction between selectivity of our PROTACs between the two isoforms of p38 versus broader consideration of the major classes of kinases is an important one. We do not wish to imply that our PROTACs show an absolute specificity for either p38 isoform: our foretinib-based PROTACs have been shown to degrade proteins beyond the p38 MAPK family – notably c-Met and the TAM family kinases – and those data were published earlier this year (Bondeson *et al.* (2018) *Cell Chem. Biol.* **25**(1): 78-87). We have now inserted a corresponding citation for these data into the text of the Introduction. Additionally, the language has been adjusted in a few spots (Introduction - 4th, 5th and final paragraphs; Discussion – 1st paragraph) to highlight the broader selectivity of foretinib and to better delineate between the ability of the degraders to unambiguously distinguish between the two isoforms of p38 from their capacity to degrade some other, non-p38 MAPK proteins of the larger kinome.

Minor concerns

“On page 5 thienopyrinone should be thienopyridinone (line 122)”

Thank you for pointing out the typographical error. We have corrected it now.

“Scheme 1: Acetona should be acetone

Thank you for pointing out the typographical error. We have corrected it now.

PRPTAC SJF-6690 should be PROTAC SJF-6690”

Thank you for pointing out the typographical error. We have corrected it now.

REVIEWERS' COMMENTS:

Reviewer #1 (Remarks to the Author):

The authors have addressed all of my concerns.

Reviewer #2 (Remarks to the Author):

In the rebuttal letter and the revised manuscript, the authors have discussed some of this reviewer's prior comments but have brushed aside the main criticism regarding a lack of dissection of the contribution of the linker. (The author response borders on being patronizing.) The authors focus on the word "ignore" in the critique. It is true that they have not ignored the linkers because they have carefully measured DC50 values for each construct. But they have not dissected the results or seek to understand the source of the results in any meaningful way. The reason this reviewer is stressing this particular point is because the authors write in the intro that they are interested in obtaining a "biophysical understanding...of the the different selectivity profiles of these PROTACs."

The crux of this reviewer's concern arises from the data listed in Table S1. Changes in DC50 values are very big for a single atom in a flexible linker. This observation begs the question: why? Such profound effect of a flexible linker on a ternary complex is perplexing and it is surprising that these results are not similarly perplexing to the authors.

The reviewer agrees that the PROTAC strategy in general is amazing, and the particular PROTAC described here for p38 is significant. However, continued advancement of PROTACs would require deeper analysis, which could be performed here. The author's response that this topic has been "covered in detail in a previous publication [Buckley et al. (2015) ACS Chem. Biol. 10(8): 1831-1837]" is unsatisfactory because as the author's know, the halotag represents a completely different scenario than the current context. One can envision, and there is data to support this, that halotag enzymes need precise positioning of the halide for enzymatic activity. The current context of protein ternary complexes can not be similarly explained.

This reviewer does not want to drag out this point if the authors are not interested in pursuing it but in this case the language in the abstract should be changed because the necessary biophysical understanding of PROTACs is not being pursued here. The paper is about p38 degradation.

Overall the manuscript is well-written and would be of interest to the community and further advance PROTACs.

Response to Reviewer #2 (Remarks to the Author) concerning initially-revised version:

“In the rebuttal letter and the revised manuscript, the authors have discussed some of this reviewer’s prior comments but have brushed aside the main criticism regarding a lack of dissection of the contribution of the linker. (The author response borders on being patronizing.) The authors focus on the word “ignore” in the critique. It is true that they have not ignored the linkers because they have carefully measured DC50 values for each construct. But they have not dissected the results or seek to understand the source of the results in any meaningful way. The reason this reviewer is stressing this particular point is because the authors write in the intro that they are interested in obtaining a “biophysical understanding...of the the different selectivity profiles of these PROTACs.”

We acknowledge Reviewer 2’s point about our having described the study as presenting a “biophysical” understanding of the phenomenon to have been insufficient as pertains to the linkers of the two PROTAC series. The pharmacological testing of the degradation by both series of p38-targeting PROTACs serves to document the threshold number of linker atoms necessary to establish or enhance selectivity, but it does not provide a biophysical, mechanistic explanation akin to the one furnished by, for example, the MD simulation-based mutagenesis that defines p38 δ selectivity. Accordingly, we have removed the descriptor “biophysical” from this revised manuscript and have added further explanations in favor of this strong point brought up by Reviewer 2 to the Results and Discussion sections (see below).

“The crux of this reviewer’s concern arises from the data listed in Table S1. Changes in DC50 values are very big for a single atom in a flexible linker. This observation begs the question: why? Such profound effect of a flexible linker on a ternary complex is perplexing and it is surprising that these results are not similarly perplexing to the authors.”

The reviewer is correct that this result is difficult to explain mechanistically with the data collected in this study. In light of that, we may not have given the observation itself its proper due in the article. We have added text to the Results section to highlight how the addition (or loss) of a single atom from the linker for the amide (or phenyl) series profoundly impacts isoform selectivity.

“In certain instances, the change in linker length by even a single carbon atom alters the degradation selectivity observed to a striking degree. For example, while SJF-8240 of the ‘amide series’ with its 12-atom linker displayed sub-micromolar potency to degrade both p38 isoforms, lengthening the linker by 1 atom to create SJF α increased potency to degrade p38 α into the low nanomolar range while simultaneously decreasing p38 δ degradation potency to the micromolar range (dramatically reducing efficacy as well). Likewise, SJF δ – of the ‘phenyl series’ with its 10 atom linker – degrades p38 δ to near-completion but is limited in degrading p38 α ; by comparison, SJF-6677, with an additional carbon in its linker degrades less than 50% of either isoform at maximum effectiveness.”

In addition, we have amended the Discussion to revisit the puzzling nature of how this selectivity is acquired by single atom differential. We have provided the following Discussion paragraph along with our changes (in red) to highlight our focus on how single atom linker changes can result in drastic degradation outcomes.

“Through our work, we demonstrated that single atom alterations to PROTAC linkers could change chemical probes with dual p38 α /p38 δ degradation activity (SJF-6693 and 6690) into ones with enhanced selectivity for the p38 α isoform (SJF-8240 and SJF α) (Supplementary Figure 2, Supplementary Table 1). These amide series PROTACs were based on the oft-utilized “left-hand side” VHL-recruiting moiety and required longer linker lengths to achieve p38 α selectivity (SJF α = 13-atom linker). Conversely, only the shortest linker-containing PROTAC of the phenyl series (SJF δ) – which employ a “right-hand side” VHL ligand linkage – demonstrated potent and efficacious activity towards a specific p38 isoform, p38 δ . Linker length increases beyond 10 atoms – even single atom changes – in the phenyl series strongly attenuated their enhanced capacity to degrade p38 δ . While it is possible that VHL might possess an inherent preference for interfacing with a given p38 MAPK isoform, we believe such differences to be nominal due to (i) high structural conservation between the p38 isoforms (Supplementary Figure 4), and (ii) the fact that the shorter amide series PROTACs (10-atom and 11-atom) show similar efficacy to degrade p38 α and p38 δ . Instead, we believe that these atomic linker length preferences are biased by the VHL-recruiting ligands themselves and selectivity is further enhanced by linker length exploration. Based on the two linkage vectors of the VHL ligand (Fig. 1a-b), we believe that the phenyl attachment provides a more direct VHL recruitment that does not require the PROTAC linker to bend back on itself, as seen in the only PROTAC crystal structure to date (PDB: 5T35 [<http://dx.doi.org/10.2210/pdb5T35/pdb>]) which is based on an amide attachment to the VHL ligand³⁶. This kinked linker conformation was also seen in our previous study in which MD simulation of the p38 α :PROTAC:VHL ternary complex revealed a residue on p38 α (Ala40) that favorably interacted with the amide series PROTAC linker³⁷. While these kinked linkers seen in the amide-linked VHL PROTACs appear to provide additional surface for favorable contact between the substrate:ligase interface, subtle changes to these contacts – either through slight modifications in linker length or composition – appear to result in drastic enhancements/reductions in potency and selectivity (as seen in Supplementary Figure 2, Supplementary Table 1). In fact, the lack of degradation seen with phenyl-linked VHL PROTACs in a previous report⁶³ might be a consequence of underexplored linker space and the fact that amide-linked and phenyl-linked VHL PROTACs can have different linker length requirements per given substrate. Thus, previous work and the current study both demonstrate that PROTAC linkers represent a delicate balance between affinity contributions and steric effects and how single atom changes to the linker can drastically shift that balance.”

The reviewer agrees that the PROTAC strategy in general is amazing, and the particular PROTAC described here for p38 is significant. However, continued advancement of PROTACs would require deeper analysis, which could be performed here. The author's response that this topic has been “covered in detail in a previous publication [Buckley et al. (2015) ACS Chem. Biol. 10(8): 1831-1837]” is unsatisfactory because as the author's know, the halotag represents a completely different scenario than the current context. One can envision, and there is data to support this, that halotag enzymes need precise positioning of the halide for enzymatic activity. The current context of protein ternary complexes can not be similarly explained.

This reviewer does not want to drag out this point if the authors are not interested in pursuing it but in this case the language in the abstract should be changed because the necessary biophysical understanding of PROTACs is not being pursued here. The paper is about p38 degradation.

Overall the manuscript is well-written and would be of interest to the community and further advance PROTACs.

The Reviewer is correct in stating that the biophysical specifics underlying the single PEG unit-dependent changes in activity for the HaloPROTACs are unlikely to explain the cognate differences seen with our p38-targeting PROTACs: the differences in PPI amino acids and three-dimensional protein interfaces between HaloTag and VHL vs. p38 and VHL would preclude extrapolation of anything meaningful from the former to the latter in terms of mechanism. Our intention in raising the example of the HaloPROTACs was merely to point out that it has been observed previously that small changes in the number of linker atoms can result in large functional changes, which made the same finding in this article a little less surprising to us. Certainly, giving more attention to this interesting phenomenon here is worthwhile, so we thank the Reviewer for bringing this point up again.

However, we do agree with Reviewer 2 that this study will not benefit from excessive deliberation on this point. To this end, we have revised the text of the Results and Discussion to elaborate on the functional differences observed from single atom linker length differences among the PROTAC series and removed the adjective “biophysical” from the article Abstract and Discussion. It is our hope that these changes will raise greater attention to this particular finding of the article, spur further research into PROTAC ‘linkerology’, as well as make the language of the article more consistent with the experimental methods and data included therein.